# ECG Representation Learning with Multi-Modal EHR Data

**Sravan Kumar Lalam**[✉,1]     **Hari Krishna Kunderu**[1]     **Shayan Ghosh**[1]     **Harish Kumar A**[1]

**Ashim Prasad**[1]     **Francisco Lopez-Jimenez**[3]     **Samir Awasthi**[1,2]     **Zachi I Attia**[3]

**Samuel J. Asirvatham**[3]     **Paul A. Friedman**[3]     **Rakesh Barve**[1,2]     **Melwin Babu**[✉,1]

[1] *Nference Inc.*     [2] *Anumana Inc.*     [3] *Mayo Clinic, USA*

[✉]Corresponding authors: {sravankumar.l@nference.net, melwin@nference.net}

**Reviewed on OpenReview:** *https://openreview.net/forum?id=UxmvCwuTMG*

## Abstract

Electronic Health Records (EHRs) provide a rich source of medical information across different modalities such as electrocardiograms (ECG), structured EHRs (sEHR), and unstructured EHRs (text). Inspired by the fact that many cardiac and non-cardiac diseases influence the behavior of the ECG, we leverage structured EHRs and unstructured EHRs from multiple sources by pairing with ECGs and propose a set of three new multi-modal contrastive learning models that combine ECG, sEHR, and text modalities. The performance of these models is compared against different baseline models such as supervised learning models trained from scratch with random weights initialization, and self-supervised learning models trained only on ECGs. We pre-train the models on a large proprietary dataset of about 9 *million* ECGs from around 2.4 *million* patients and evaluate the pre-trained models on various downstream tasks such as classification, zero-shot retrieval, and out-of-distribution detection involving the prediction of various heart conditions using ECG waveforms as input, and demonstrate that the models presented in this work show significant improvements compared to all baseline modes.

## 1 Introduction

Electronic Health Records (EHRs) are generated for every patient encounter or event and are becoming increasingly available in recent years. These are multi-modal in nature and capture rich phenotypic information of the patients over time in the form of structured EHRs and unstructured EHRs. Structured EHRs (denoted sEHR) contain information about diagnoses, procedures, medication prescriptions, lab tests, vitals, and more, while unstructured EHRs encompass clinical notes, radiology images, pathology images, echocardiogram videos, time series ECG signals, etc. Recently multi-modal contrastive learning methods applied to radiology and pathology images by pairing with the corresponding medical reports to learn medical image representations (Zhang et al., 2022; Huang et al., 2021; Boecking et al., 2022; Bannur et al., 2023; Lu et al., 2023) have shown promising results on downstream tasks such as classification, image-text retrieval, etc. These methods generally have two stages: (i) In stage I, the model is pre-trained on large unlabelled data to learn generic representations by maximizing the alignment between embeddings of different modalities in latent space. (ii) In stage II, the model is fine-tuned on a task-specific labeled dataset by transferring the knowledge from the pre-trained model. However, ECG representation learning by pairing with EHRs via multi-modal contrastive learning is less explored. Uni-modal contrastive learning similar to Chen et al. (2020a) has been applied to the ECG domain to learn ECG representations (Kiyasseh et al., 2021; Diamant et al., 2022; Gopal et al., 2021; Mehari & Strodthoff, 2022; Oh et al., 2022), but they lack the ability to compare different modalities in latent space using similarity metrics like cosine similarity for use in zero-shot transfer learning. Also, contrastive learning using multi-modal data produces high-quality representations as they exploit information from multiple sources and extract semantics by aligning with various modalities.

ECG is a simple, non-invasive test that records the electrical activity of the heart and is helpful in diagnosing heart conditions and patient monitoring. In recent years, deep learning techniques have been employed on ECG data to predict various heart conditions, even in cases where diagnostic criteria using ECGs have not been firmly established in clinical practice (Tison et al., 2019; Hannun et al., 2019; Galloway et al., 2019; Attia et al., 2019a;b;c; Ko et al., 2020; Adedinsewo et al., 2020; Christopoulos et al., 2020; Yao et al., 2021; Siontis et al., 2021; Cohen-Shelly et al., 2021; Bos et al., 2021; Grogan et al., 2021; Ahn et al., 2022). Gopal et al. (2021) discusses that supervised learning models of this nature demand extensive, high-quality datasets with precise annotations to achieve robust generalization on real-world data. Unfortunately, within the healthcare domain, acquiring such labeled datasets is challenging, as they are scarce, expensive, and time-consuming to obtain due to the necessity of trained physicians for the annotations. Motivated by the following facts: (i) multi-modal contrastive learning applied to the general domain images (Radford et al., 2021; Jia et al., 2021; Goel et al., 2022) as well as the medical domain images (Zhang et al., 2022; Huang et al., 2021; Boecking et al., 2022; Bannur et al., 2023; Lu et al., 2023) by pairing images with text data has demonstrated promising results; (ii) ECG signals contain information related to both cardiovascular and non-cardiovascular diseases (Venn et al., 2022); (iii) EHR data capture rich phenotypic information of the patients over time, we address the challenges described previously by leveraging EHRs. We pair structured EHR and unstructured EHR data with ECGs to learn ECG representations via multi-modal contrastive learning. In particular, we utilize diagnosis codes, procedure codes, and medication prescriptions from the structured EHR category and text data from various sources such as ECG reports, ECHO reports, radiology reports, pathology reports, microbiology reports, clinical notes, and surgical notes from the unstructured EHR category. Our contributions are summarised as follows:

1. We propose **sEHR-BERT**, a BERT model pre-trained to encode sEHR modality for use in multi-modal contrastive learning models.

2. We propose a set of three multi-modal contrastive learning models that combine sEHR, ECG, and text modalities to learn ECG representations:

   - **ECG-sEHR**: A model that combines ECG and sEHR modalities,
   - **ECG-Text**: A model that combines ECG and text modalities,
   - **sEHR-ECG-Text**: A model that combines sEHR, ECG, and text modalities.

3. We then compare the effectiveness of the pre-trained models on downstream tasks such as linear classification, fine-tuning, zero-shot retrieval, and out-of-distribution detection with different baseline models including supervised learning models trained from scratch with random initialization and current state-of-the-art (SOTA) ECG-only self-supervised learning models.

## 2 Related Work

### 2.1 Contrastive Learning for General Domain Images

Self-supervised learning (SSL) using contrastive learning methods has emerged as a powerful pre-training technique to learn generic representations of the data. These methods learn representations either (i) by pulling the embeddings of similar pairs (positive pairs) together and pushing the embeddings of dissimilar pairs (negative pairs) apart in the latent embedding space or (ii) by contrasting cluster assignments. Some of the notable works in computer vision include InfoNCE (Oord et al., 2018), SimCLR (Chen et al., 2020a), SimCLRv2 (Chen et al., 2020b), MoCo (He et al., 2020), SupCon (Khosla et al., 2020), SEER (Goyal et al., 2021), PIRL (Misra & Maaten, 2020), SwAV (Caron et al., 2020), and PCL (Li et al., 2021). These methods come under the category of uni-modal contrastive learning as they utilize only one type of data modality, i.e., images. Multi-modal contrastive learning by pairing general domain images with the corresponding image captions to learn image-text embeddings jointly in the shared space (Radford et al., 2021; Jia et al., 2021; Goel et al., 2022) has shown impressive results on downstream tasks such as zero-/few-shot learning.

## 2.2 Contrastive Learning for Medical Domain Images

Uni-modal contrastive learning based on SimCLR has been applied to medical domain images (Azizi et al., 2021; 2022; Ciga et al., 2022; Wang et al., 2022; Srinidhi & Martel, 2021; Sowrirajan et al., 2021) to learn medical image representations. Motivated by some of the initial works in uni-modal contrastive learning, ConVIRT (Zhang et al., 2022) proposed a multi-modal contrastive learning method by pairing chest radiology images with the corresponding radiology reports. Huang et al. (2021) extended ConVIRT for learning local and global representations by contrasting image sub-regions with words in the medical report. Boecking et al. (2022); Bannur et al. (2023) made improvements in the radiology domain by leveraging longitudinal medical images, building a domain-specific language model for radiology reports, and adding Masked Language Modeling (MLM) loss to contrastive loss during joint vision-language pre-training. Lu et al. (2023) applied multi-modal contrastive learning by pairing histopathology whole slide images with pathology reports. Our multi-modal contrastive learning models are largely inspired by ConVIRT (Zhang et al., 2022).

## 2.3 Contrastive Learning for Time Series ECG signals

SimCLR and the other aforementioned uni-modal contrastive learning models were developed for use in computer vision. However, they have been adopted for use with time series ECG signals in subsequent works (Kiyasseh et al., 2021; Diamant et al., 2022; Gopal et al., 2021; Mehari & Strodthoff, 2022; Oh et al., 2022). The principle difference between CLOCS (Kiyasseh et al., 2021), PCLR (Diamant et al., 2022) and the 3KG (Gopal et al., 2021) models is in the way the positive pairs are created. CLOCS treats consecutive non-overlapping segments and/or leads of the same ECG as positive pairs. PCLR treats two ECGs of the same patient as positive pairs. 3KG constructs positive pairs by applying spatial augmentations such as rotation and scaling in vectorcardiogram (VCG) space after converting ECG to VCG, followed by temporal augmentations such as time masking in ECG space after converting VCG back to ECG. Cheng et al. (2020) introduced adversarial training to address intersubject variability while learning ECG representations using contrastive learning. Mehari & Strodthoff (2022) adapted SimCLR (Chen et al., 2020a), CPC (Oord et al., 2018), and SwAV (Caron et al., 2020) to ECG domain to learn ECG representations. Recently, Oh et al. (2022) proposed a pre-training method that combines CMSC from Kiyasseh et al. (2021) and Wave2Vec 2.0 (Baevski et al., 2020) from speech domain to learn local and global contextual ECG representations.

To the best of our knowledge, there is only one work that combines ECGs with other modalities. Raghu et al. (2022) developed a SimCLR-like contrastive learning model that was pre-trained using multi-modal clinical time series data such as ECG signals and structured time series data (labs and vitals). The model utilizes 18-dimensional structured time series data from metabolic panel, blood pressures, heart rate, and SpO2. The model is shown to have achieved improved or comparable performance over training from scratch on two downstream tasks: (i) Elevated mPAP and (ii) 24-hour mortality rate. To the best of our knowledge, we are the first to fully utilize the large landscape of electronic health records to learn ECG representations.

# 3 Methods

## 3.1 sEHR-BERT: Structured EHR Model Pre-training

Several methods have been proposed to model structured EHRs based on BERT (Devlin et al., 2018): BEHRT (Li et al., 2020), Med-BERT (Rasmy et al., 2021), CEHR-BERT (Pang et al., 2021), and CEHR-GAN-BERT (Poulain et al., 2022). However, none of these pre-trained models are publicly available to use in our work. Moreover, the vocabulary in our dataset may not be aligned with the vocabulary of the mentioned models. So we developed sEHR-BERT, a model pre-trained to encode sEHR data and produce sEHR representations based on the BERT architecture (Devlin et al., 2018). We used a vocabulary of size 28593, constructed from ICD diagnosis codes, ICD procedure codes, and medication prescriptions. These are collectively referred to as "medical codes" in this work. The input to the model is a sequence of medical codes sorted in ascending order based on medical codes' timestamps. Each code is processed by adding its corresponding medical code embedding, time embedding, and medical code type embedding and sent to the transformer encoder. Time embeddings are constructed so that codes falling within non-overlapping 7-day windows share a common embedding. Medical code type embeddings are divided into different categories

(i.e., diseases, symptoms, procedures, special tokens, etc.). We used a custom BERT model with the number of layers, hidden size, and number of self-attention heads set to 5, 320, and 5 respectively. This model has 15M parameters. We initialize the model weights randomly and follow the BERT (Devlin et al., 2018) pre-training strategy, i.e., Masked Language Modeling (MLM) to learn the representations of the structured EHR sequences. We minimize the MLM loss given by $\mathcal{L} = -\frac{1}{K} \sum_{i=1}^{K} \log p\left(D_{m_i} | D_{\bar{M}}; \Theta\right)$, where $\Theta$ are parameters of the model, $D = \{D_0, D_1, ..., D_N\}$ is the sequence of medical codes of length N, $M = \{m_0, m_1, ..., m_K\}$ are indices of masked medical codes, and $D_{\bar{M}}$ denotes the set of unmasked medical codes. During training, the medical codes are masked with a probability of 15%, and the model is trained with AdamW (Loshchilov & Hutter, 2019) optimizer and batch size of 512 for 100 epochs. We set an initial learning rate of 5e-4 and the learning rate is reduced by a factor of 2 if the validation loss stops decreasing continuously for 2 epochs.

### 3.2 sEHR-ECG-Text: Joint sEHR, ECG and Text Pre-training

In this section, we describe the pre-training of the sEHR-ECG-Text model, where we pair the ECG modality with sEHR and text modalities to jointly learn multi-modal representations.

#### 3.2.1 Preliminaries

MultiModal Versatile Networks (MMV) (Alayrac et al., 2020) apply contrastive learning to multi-modal data including video, audio, and text under the assumption that the video and audio modalities are more granular than the text modality. MMV discusses that applying contrastive loss in *shared space* (where all modalities are embedded into a single shared vector space) may not maintain specificities, as it implicitly assumes that all modalities have equal granularity. To address this, MMV proposes to learn two separate embedding spaces: a fine-grained space where video and audio are matched and a coarse-grained space where text is matched with video and audio domains. This method is referred to as *fine and coarse spaces (FAC)*. We hypothesize that sEHR, ECG, and text modalities do not exhibit equal granularity. Moreover, the ECGs are paired with sEHR and text data within a specific time window surrounding the ECG acquisition timestamp, and tokens are trimmed based on the input length accepted by the corresponding encoders, as we describe in Sections 4.2.2 and 4.2.3. This implies that the same level of information might not be maintained between sEHR and text, so we adopt the FAC framework from MMV in our sEHR-ECG-Text model. We describe the methodology in detail in the following sections.

#### 3.2.2 Notation

Let $x \in \mathcal{X}$ be an instance defined by an instantiation of different modalities $\mathcal{M} : x = \{x_m\}$, $m \in \mathcal{M}$. In this study, we employ three modalities: ECG $x_e \in \mathcal{X}_e$, sEHR $x_s \in \mathcal{X}_s$, and text $x_t \in \mathcal{X}_t$. Specifically, $x_s$, $x_e$, and $x_t$ represent the sequence of sEHR codes, ECG waveform samples, and sequence of text tokens respectively. Let $E_m : \mathcal{X}_m \to \mathbb{R}^{d_m}$ be a parameterized modality-specific encoder that takes as input an instance $x_m$ from modality $m$ and produces a modality-specific representation of dimension $d_m$. These modality-specific representations are embedded into a shared space $\Omega_z \subset \mathbb{R}^{d_z}$, where $z$ represents the list of modalities that we embed into this space. For instance, $z = es$ to denote joint ECG-sEHR space $\Omega_{es}$, $z = et$ to denote joint ECG-Text space $\Omega_{et}$, or $z = set$ to denote joint sEHR-ECG-Text space $\Omega_{set}$. In this shared space, we maximize or minimize the alignment between different modalities using the contrastive loss objective. The projection head $P_{m \to z} : \mathbb{R}^{d_m} \to \mathbb{R}^{d_z}$ is used to embed modality specific-representations $v_m = E_m(x_m)$ into the shared space $\Omega_z$, and we denote the resulting vector as $v_{m,z} = P_{m \to z}(E_m(x_m))$, which signifies the representation of the input modality $x_m$ in the shared space $\Omega_z$.

#### 3.2.3 Data Encoding

To obtain the modality-specific representations, we use a convolutional neural network (CNN) customized to one dimension ($E_e$) for the ECG modality, the pre-trained sEHR-BERT ($E_s$) as described in Section 3.1 for the sEHR modality, and pre-trained GatorTron (Yang et al., 2022) ($E_t$) for the text modality. Global average pooling is applied at the final layer for all three encoders to obtain the representations. We use multi-layer perceptron (MLP) for the projection heads.

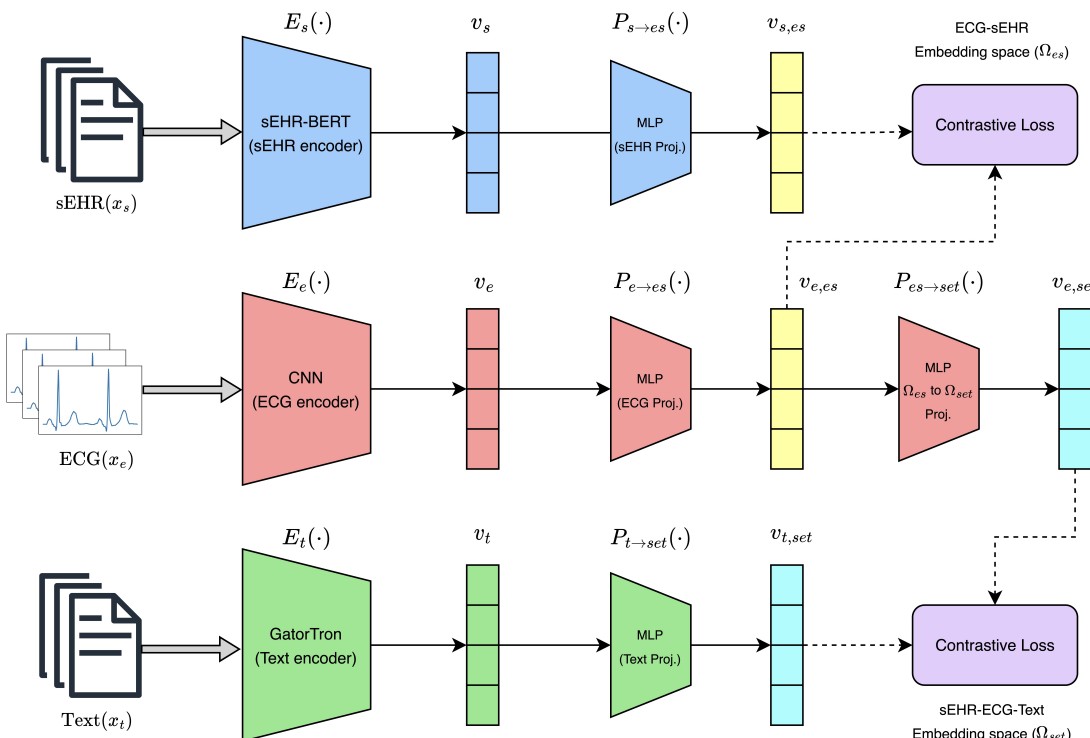

Figure 1: Overview of the sEHR-ECG-Text model pre-training. The model takes as input three modalities $\mathcal{M} : x = \{x_m\}$, $m \in \mathcal{M}$, i.e., ECG ($x_e$), sEHR ($x_s$), and text ($x_t$). $E_m(\cdot)$ and $v_m$ denote the modality-specific encoder and modality-specific representation respectively. $\Omega_z$ denotes the shared embedding space, where $z$ represents the list of modalities that we embed into this space. For instance, $z = es$ to denote joint ECG-sEHR space $\Omega_{es}$. $P_{m \to z}(\cdot)$ denotes the projection head used to embed modality specific representation $v_m$ into shared space $\Omega_z$. $v_{m,z}$ denotes the representation of the input modality $x_m$ in the shared space $\Omega_z$. The model is trained by applying contrastive loss between ECG and sEHR in the fine-grained ECG-sEHR space ($\Omega_{es}$) and between ECG and text in the coarse-grained sEHR-ECG-Text space ($\Omega_{set}$).

### 3.2.4 Multi-Modal Contrastive Objective

As mentioned before, we use FAC framework inspired from MMV (Alayrac et al., 2020) where ECG and sEHR are compared in fine-grained joint ECG-sEHR space $\Omega_{es}$, while ECG is compared with text in coarse-grained joint sEHR-ECG-Text space $\Omega_{set}$. Given a minibatch containing N instances $\{x^i\}_{i=1}^N$, we denote $v_{m,z}^i = P_{m \to z}(E_m(x_m^i))$ as the representation of the modality $m$ in the shared space $\Omega_z$ for the $i$-th instance. Following Zhang et al. (2022), we define the contrastive objective bidirectionally. For example, in the case of contrastive loss between ECG and sEHR domains, the loss is directed from ECG to sEHR and vice-versa. In the context of contrastive objective, we consider N pairs of ECG-sEHR ($x_e, x_s$) as positive, while the remaining $N^2 - N$ pairs are treated as negative. The same approach is applied to contrastive loss between ECG and text domains. Let $sim(x, y) = x^T y / \|x\| \|y\|$ denote the cosine similarity between two vectors $x, y \in \mathbb{R}^{d_z}$, $\mathcal{L}_{es}$ be the contrastive loss between ECG and sEHR, $\mathcal{L}_{et}$ be the contrastive loss between ECG and text, $\lambda_{es}$ and $\lambda_{et}$ be scalar weights $\in [0, 1]$, and $\tau \in \mathbb{R}^+$ be the temperature parameter. The combination of $\lambda_{es}$ and $\lambda_{et}$ gives the overall loss, denoted by $\mathcal{L}$ (Equation 3), and we aim to minimize this loss.

$$\mathcal{L}_{es} = -\frac{1}{N} \sum_{i=1}^N \left( \lambda_{es} \log \frac{\exp\left(sim(v_{e,es}^i, v_{s,es}^i)/\tau\right)}{\sum_{k=1}^N \exp\left(sim(v_{e,es}^i, v_{s,es}^k)/\tau\right)} + (1 - \lambda_{es}) \log \frac{\exp\left(sim(v_{s,es}^i, v_{e,es}^i)/\tau\right)}{\sum_{k=1}^N \exp\left(sim(v_{s,es}^i, v_{e,es}^k)/\tau\right)} \right) \quad (1)$$

$$\mathcal{L}_{et} = -\frac{1}{N} \sum_{i=1}^{N} \left( \lambda_{et} \log \frac{\exp\left(sim(v^i_{e,set}, v^i_{t,set})/\tau\right)}{\sum_{k=1}^{N} \exp\left(sim(v^i_{e,set}, v^k_{t,set})/\tau\right)} + (1 - \lambda_{et}) \log \frac{\exp\left(sim(v^i_{t,set}, v^i_{e,set})/\tau\right)}{\sum_{k=1}^{N} \exp\left(sim(v^i_{t,set}, v^k_{e,set})/\tau\right)} \right) \tag{2}$$

$$\mathcal{L} = \mathcal{L}_{es} + \mathcal{L}_{et} \tag{3}$$

Figure 1 illustrates the pre-training of the sEHR-ECG-Text model. See Appendix B.1 for more details about implementation and training. We also present *shared space* versus *FAC spaces* ablation in Appendix E. We provide the architecture of the shared space sEHR-ECG-Text model in Appendix B.2.

### 3.3 ECG-sEHR: Joint ECG and sEHR Pre-training

In the ECG-sEHR model, we pair ECG signals with structured EHRs as we describe in more detail in Sections 4.2.1 and 4.2.2. We apply contrastive objective between ECG and sEHR modalities in joint ECG-sEHR embedding space ($\Omega_{es}$), where we minimize the contrastive loss given in Equation 1. We provide the ECG-sEHR model architecture in Appendix B.2.

### 3.4 ECG-Text: Joint ECG and Text Pre-training

In the ECG-Text model, we pair ECG signals with clinical text data from unstructured EHRs as we describe in more detail in Section 4.2.3. ECG and text embeddings are jointly learned by applying the contrastive objective between ECG and text modalities in joint ECG-Text embedding space ($\Omega_{et}$). We provide the ECG-Text model architecture in Appendix B.2. Following the notation introduced in Section 3.2, we minimize the contrastive loss given by,

$$\mathcal{L}_{et} = -\frac{1}{N} \sum_{i=1}^{N} \left( \lambda_{et} \log \frac{\exp\left(sim(v^i_{e,et}, v^i_{t,et})/\tau\right)}{\sum_{k=1}^{N} \exp\left(sim(v^i_{e,et}, v^k_{t,et})/\tau\right)} + (1 - \lambda_{et}) \log \frac{\exp\left(sim(v^i_{t,et}, v^i_{e,et})/\tau\right)}{\sum_{k=1}^{N} \exp\left(sim(v^i_{t,et}, v^k_{e,et})/\tau\right)} \right) \tag{4}$$

In this model, note that ECG and text are compared in joint ECG-Text space ($\Omega_{et}$), which differs from the sEHR-ECG-Text model where ECG and text are compared in the sEHR-ECG-Text space ($\Omega_{set}$).

## 4 Experiments and Results

### 4.1 Dataset Splits Setup

We used EHR data of around 2.4 *million* patients from Mayo Clinic, USA consisting of around 9 *million* ECGs to create datasets for pre-training and downstream tasks. These EHRs have undergone a rigorous de-identification process, guaranteeing the utmost privacy and data security. These records have received approval from the Institutional Review Board (IRB) of Mayo Clinic, USA, ensuring compliance with ethical guidelines and regulations. Consequently, there are no privacy or data security issues associated with the use of these de-identified EHRs. We initially split all the patients into the global train, validation, and test sets in a 60%, 5%, and 35% ratio, which are then used to create pre-training datasets and disease cohorts for downstream classification tasks. In particular train, validation, and test sets for pre-training and classification tasks are created by drawing the EHRs from the global train, validation, and test patients respectively. This approach ensures that we can effectively evaluate the quality of representations on downstream tasks, as the data of validation and test patients is not seen during the pre-training phase. Consequently, all datasets across tasks have train, validation, and test split percentages roughly close to 60%, 5%, and 35% respectively.

## 4.2 Pre-training Datasets

In this section, we describe the creation of following datasets for pre-training the proposed models: (i) sEHR sequences for pre-training the sEHR-BERT model, (ii) ECG-sEHR $(x_e, x_s)$ pairs for pre-training the ECG-sEHR model, and (iii) ECG-Text $(x_e, x_t)$ pairs for pre-training the ECG-Text model. For sEHR-ECG-Text model pre-training, we construct triplets $(x_s, x_e, x_t)$ by considering $(x_e, x_s)$ and $(x_e, x_t)$ pairs that have ECG paired with both sEHR and text data. See Appendix A.1 for details on the number of instances and patients used in different pre-training models.

### 4.2.1 Dataset for sEHR-BERT Pre-training

We use ICD-9 (International Classification of Diseases, Ninth Revision), ICD-10 (International Classification of Diseases, Tenth Revision), CPT (Current Procedural Terminology), HCPS (Healthcare Common Procedures Coding System) codes, and medication prescriptions to create the dataset for sEHR-BERT pre-training. Since ICD-9 codes differ from ICD-10 codes, but their corresponding text descriptions are similar, we map ICD-9 to ICD-10 to maintain consistent phenotypic information. ICD-10 diagnosis codes are shortened to the first three characters as keeping four or more characters provides little to no extra information for large-scale pre-training. For example, the corresponding text descriptions of ICD-10 diagnosis codes I26.0 and I26.9 are *pulmonary embolism with acute cor pulmonale* and *pulmonary embolism without acute cor pulmonale* respectively, but these come under a common disease category, i.e., *pulmonary embolism* (I26). Shortened ICD-10 diagnosis codes, ICD-10 procedure codes, CPT codes, HCPS codes, and medication prescriptions associated with at least 50 patients are included in the vocabulary, resulting in a size of 28593. We present short ICD-10 vs. full ICD-10 diagnosis codes ablation while keeping codes from other sources consistent in Appendix E. To create the sEHR sequence for sEHR-BERT model pre-training we randomly select one sequence of up to 512 consecutive medical codes from a given patient's timeline. On average, the sequence length of the resulting dataset is 168.

### 4.2.2 ECG-sEHR Pairs Generation

To create the ECG-sEHR $(x_e, x_s)$ pairs, we first select an ECG of a given patient, $x_e$, and consider all the shortened ICD-10 diagnosis codes, ICD-10 procedure codes, CPT codes, HCPS codes, and medication prescriptions associated with that patient within a period of one year prior, and one year subsequent, to the acquisition timestamp of that ECG. The medical codes restricted to this time range are arranged sequentially to form the sEHR input sequence $x_s$. The average length of the constructed sequences is 121.

### 4.2.3 ECG-Text Pairs Generation

ECGs are paired with unstructured EHR text data derived from various sources, including ECG reports, ECHO reports, pathology reports, radiology reports, microbiology reports, clinical notes, and surgical notes, collectively referred to as "patient notes" in this work. Despite the GatorTron-base (Yang et al., 2022) model's capacity to handle sequences of up to 512 tokens, we were limited to work with a maximum of 400 tokens due to computing resource constraints. Given the abundance of text data within patient notes, we implemented a filtering process to extract only the most relevant information. Specifically, we selected patient notes that contained entities from a predefined list of biomed-

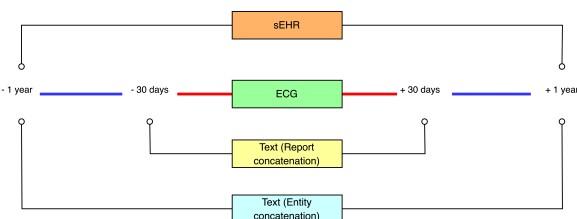

Figure 2: The figure shows the linkage of sEHR and text modalities with ECG modality within specific time windows: one year for the sEHR modality, one month for report concatenation, and one year for entity concatenation in the case of the text modality.

ical entity types, such as diseases, symptoms, procedures, medications, biomarkers, and gene mutations, using an in-house NLP model. The next step involves pairing ECGs with the selected patient notes. We employed two distinct methods for this purpose: report concatenation and entity concatenation.

**Report concatenation:** For each patient's ECG, we concatenate all selected patient notes that are available within one month around the ECG acquisition timestamp. This approach resulted in an average sequence length of 354 tokens after tokenization.

**Entity concatenation:** It's important to note that aligning ECGs with patient information spanning a more extended timeframe is advantageous for understanding ECG patterns more effectively. To effectively capture a broader spectrum of medical insights while adhering to token constraints, we concatenate only the identified entities from the patient notes within a one-year timeframe around the ECG acquisition timestamp. This resulted in an average sequence length of 266 after tokenization. Figure 2 illustrates how sEHR and text modalities are linked with ECG modality within specific time windows around the ECG timestamp.

We use entity concatenation for the main results as it yielded better results when compared to report concatenation. We also present the report concatenation vs. entity concatenation ablation in Appendix E.

### 4.3 Classification Datasets

In this section, we provide the details of the classification datasets that are used in linear classification and fine-tuning tasks. We evaluate the pre-trained models on both internal and external datasets.

#### 4.3.1 Internal Datasets

We target six cardiac diseases whose diagnostic criteria using ECGs haven't been established in clinical practice, i.e., either the patterns to identify these diseases from ECG are not known or ECG is not the gold standard for definitive diagnosis. These include coronary atherosclerosis, myocarditis, cardiac amyloidosis, pulmonary hypertension (PH), low left ventricular ejection fraction (low LVEF, i.e., LVEF$\leq$40), and atrial fibrillation in normal sinus rhythm (AFib in NSR). These diseases are diagnosed by other means and the diagnostic information is available in EHRs. For example, to diagnose PH, an invasive, right heart catheterization (RHC) procedure is performed and to identify low LVEF, an echocardiogram test is performed. we utilize EHRs to associate ECGs with these diseases and generate labels. See Appendix A.2 for the summary of the disease labeling process, the number of ECGs, and the number of patients used.

#### 4.3.2 External Datasets

We also evaluate all pre-trained models on two publicly available datasets: (i) **PhysioNet2020** (Alday et al., 2020), which consists of a collection of six 12-lead ECG datasets with varying signal lengths and sampling rates, (ii) **Chapman** (Zheng et al., 2020), which contains 10-second long 12-lead ECGs (see Appendix A.3 for more details). The diseases in these datasets are commonly diagnosed by physicians directly using ECGs, unlike the diseases in our proprietary internal classification datasets. To replicate the state-of-the-art results on the PhysioNet2020 dataset presented by 3KG (Gopal et al., 2021), we followed their detailed procedure: (i) merge some of the conditions from the list of 27 conditions due to their similarity, i.e., complete right bundle branch block and right bundle branch block, premature atrial contraction and supraventricular premature beats, premature ventricular contractions and ventricular premature beats, 1st-degree atrioventricular block and prolonged PR interval, and evaluate on 23 distinct classes; (ii) resample the ECG signals to 500Hz; (iii) take non-overlapping 5-second long crops from each ECG record exhaustively and associate those with the label of the original record; (iv) split the dataset into 80%, 10%, and 10% for training, validation, and testing respectively; (v) train a 23-class multilabel classification model. For the Chapman dataset, in line with Kiyasseh et al. (2021), we merge 11 cardiac arrhythmia conditions of the dataset into 4 major classes and split the dataset into 60%, 20%, and 20% for training, validation, and testing respectively.

### 4.4 Baseline Models

**Random initialization models.** We train binary classification models on all individual diseases from scratch using the same ECG encoder that was used during pre-training, by randomly initializing the weights.

**ECG-only contrastive learning models.** We compare our models with the current state-of-the-art ECG-only self-supervised learning models. In particular, we compare against the 3KG (Gopal et al., 2021),

CLOCS(CMSC) (Kiyasseh et al., 2021) and PCLR (Diamant et al., 2022) models. For identical comparison, we pre-train all three models using the same global splits that we used for the multi-modal contrastive pre-training but with only ECG signal as input and we also use the same ECG encoder that we used while pre-training multi-modal contrastive learning models.

### 4.5 Downstream Tasks and Results

In this section, we evaluate the pre-trained models' transfer learning capabilities and compare them with different baseline methods on various downstream tasks such as classification, zero-shot retrieval, and out-of-distribution (OOD) detection.

#### 4.5.1 Classification Tasks

We evaluate the pre-trained models on linear classification and fine-tuning tasks. We evaluate the representation quality by extracting embeddings from the pre-trained ECG encoder by passing ECG waveform as input and training logistic regression models on various cardiovascular diseases using both internally created and publicly available datasets as outlined in Section 4.3. In fine-tuning tasks, we add a classification head (MLP) on top of the pre-trained ECG encoder and fine-tune the entire network. We compare our pre-trained models with various baseline models as described in Section 4.4. One of the most useful applications of pre-trained models is in providing downstream tasks with data efficiency, which ensures consistent performance with reduced training data. This is very valuable when a large amount of labeled data is not available due to the low prevalence of the diseases or is too expensive to procure. To demonstrate this, we create different fractions (1%, 10%, 100%) of the training set by maintaining the original disease prevalence. For low-data diseases such as coronary atherosclerosis, myocarditis, and cardiac amyloidosis, we drop 1% split due to small dataset sizes. We use AUROC and AUPRC as our evaluation metrics. We conduct each experiment using five random seeds and report the mean and standard deviation. For the 1% and 10% splits, we conducted experiments with five distinct fractional splits derived from the original training split (100%).

**Results and Discussion.** Table 1 shows the linear classification results on external datasets. Table 2 shows classification results on coronary atherosclerosis, myocarditis, and cardiac amyloidosis diseases (low-data) and Table 3 shows classification results on pulmonary hypertension, low LVEF, and AFib in NSR diseases (high-data). We observed the following findings: (i) Linear classifiers trained using representations obtained from our pre-trained models consistently outperform all baseline models by large margins across different fractions for all diseases. (ii) Fine-tuned classification models initialized with our pre-trained models' weights outperform all baseline models by a large margin in low-data environments and a small margin in high-data environments. (iii) Classification models trained with only 10% of the training data using our pre-trained models produce results that are as good as or better than those obtained by training on the entire dataset with baseline models. This demonstrates the effectiveness of our pre-trained models in achieving data efficiency. (iv) When utilizing the complete training dataset, on the linear classification task, our ECG-Text model achieves an AUROC score of 0.915 on PhysioNet2020, surpassing the SOTA performance reported in Gopal et al. (2021) (3KG) by 2.5% (0.915 vs. 0.890). Additionally, on the Chapman dataset, it achieves an AUROC score of 0.990, surpassing the SOTA performance reported in Kiyasseh et al. (2021) (CLOCS) by 3.1% (0.990 vs. 0.959). It's worth noting that, on equal grounds, we surpass 3KG by 4.3% (0.915 vs. 0.872) and CLOCS by 4.4% (0.990 vs. 0.946). (v) We hypothesize that the main reason for the poor performance of ECG-only contrastive learning models on internal datasets is their exclusive dependence on ECGs for learning representations, i.e., comparing different instances of ECG data. This may not be sufficient to learn the complex patterns of various medical conditions. In contrast, our methods align ECGs with EHR data, providing rich contextual information about a patient's health history, including diagnoses, procedures, medications, and more. This approach is beneficial for learning ECG patterns more effectively.

**Comparison between ECG-sEHR and ECG-Text models.** The ECG-sEHR model demonstrates superior performance on internal datasets, whereas the ECG-Text model excels on external datasets. We hypothesize that this difference arises from the fact that external datasets, such as PhysioNet2020 and Chapman, primarily comprise diseases that are commonly diagnosed from ECGs (i.e., arrhythmias, conduction

Table 1: Linear classification results (AUROC, mean and standard deviation over 5 runs with different random seeds) for PhysioNet2020 (23 classes) and Chapman (4 classes) datasets. Results within 95% confidence intervals of the best result are shown in **bold**.

| Method | PhysioNet2020 | | | Chapman | | |
|---|---|---|---|---|---|---|
| | 1% | 10% | 100% | 1% | 10% | 100% |
| *Supervised baseline* | | | | | | |
| Random Init. | $0.684 \pm 0.024$ | $0.842 \pm 0.006$ | $0.913 \pm 0.006$ | $0.552 \pm 0.031$ | $0.964 \pm 0.003$ | $0.987 \pm 0.002$ |
| *ECG-only SSL* | | | | | | |
| 3KG | $0.774 \pm 0.001$ | $0.837 \pm 0.003$ | $0.872 \pm 0.000$ | $0.934 \pm 0.008$ | $0.968 \pm 0.001$ | $0.984 \pm 0.000$ |
| CLOCS(CMSC) | $0.770 \pm 0.005$ | $0.836 \pm 0.002$ | $0.864 \pm 0.000$ | $0.876 \pm 0.019$ | $0.936 \pm 0.001$ | $0.946 \pm 0.000$ |
| PCLR | $0.721 \pm 0.003$ | $0.798 \pm 0.003$ | $0.834 \pm 0.001$ | $0.722 \pm 0.007$ | $0.821 \pm 0.004$ | $0.872 \pm 0.000$ |
| *Our models* | | | | | | |
| sEHR-ECG-Text | $0.813 \pm 0.005$ | $0.882 \pm 0.003$ | $0.911 \pm 0.000$ | $0.957 \pm 0.004$ | $0.979 \pm 0.001$ | $0.985 \pm 0.000$ |
| ECG-sEHR | $0.788 \pm 0.005$ | $0.866 \pm 0.002$ | $0.896 \pm 0.000$ | $0.937 \pm 0.010$ | $0.969 \pm 0.000$ | $0.976 \pm 0.000$ |
| ECG-Text | $\mathbf{0.820 \pm 0.003}$ | $\mathbf{0.887 \pm 0.001}$ | $\mathbf{0.915 \pm 0.000}$ | $\mathbf{0.974 \pm 0.002}$ | $\mathbf{0.987 \pm 0.001}$ | $\mathbf{0.990 \pm 0.000}$ |

Table 2: Results (AUROC, mean and standard deviation over 5 runs with different random seeds) for coronary atherosclerosis, myocarditis, and cardiac amyloidosis classification tasks: (a) linear classification, (b) fine-tuned classification. Results within 95% confidence intervals of the best result are shown in **bold**. See Appendix D for AUPRC metrics.

(a) Linear classification

| Method | Coronary atherosclerosis | | Myocarditis | | Cardiac amyloidosis | |
|---|---|---|---|---|---|---|
| | 10% | 100% | 10% | 100% | 10% | 100% |
| *Supervised baseline* | | | | | | |
| Random Init. | $0.772 \pm 0.012$ | $0.831 \pm 0.004$ | $0.775 \pm 0.019$ | $0.868 \pm 0.003$ | $0.912 \pm 0.004$ | $0.945 \pm 0.001$ |
| *ECG-only SSL* | | | | | | |
| 3KG | $0.722 \pm 0.008$ | $0.786 \pm 0.000$ | $0.699 \pm 0.022$ | $0.808 \pm 0.000$ | $0.869 \pm 0.009$ | $0.906 \pm 0.000$ |
| CLOCS(CMSC) | $0.743 \pm 0.004$ | $0.801 \pm 0.000$ | $0.662 \pm 0.021$ | $0.783 \pm 0.000$ | $0.876 \pm 0.006$ | $0.918 \pm 0.000$ |
| PCLR | $0.761 \pm 0.004$ | $0.825 \pm 0.000$ | $0.718 \pm 0.022$ | $0.818 \pm 0.000$ | $0.898 \pm 0.006$ | $0.928 \pm 0.000$ |
| *Our models* | | | | | | |
| sEHR-ECG-Text | $\mathbf{0.840 \pm 0.003}$ | $0.890 \pm 0.000$ | $\mathbf{0.860 \pm 0.019}$ | $\mathbf{0.901 \pm 0.001}$ | $\mathbf{0.934 \pm 0.007}$ | $\mathbf{0.960 \pm 0.000}$ |
| ECG-sEHR | $\mathbf{0.836 \pm 0.011}$ | $\mathbf{0.891 \pm 0.000}$ | $\mathbf{0.859 \pm 0.011}$ | $0.896 \pm 0.000$ | $\mathbf{0.932 \pm 0.006}$ | $0.959 \pm 0.000$ |
| ECG-Text | $0.821 \pm 0.006$ | $0.875 \pm 0.000$ | $0.785 \pm 0.010$ | $0.881 \pm 0.000$ | $0.922 \pm 0.006$ | $0.949 \pm 0.000$ |

(b) Fine-tuned classification

| Method | Coronary atherosclerosis | | Myocarditis | | Cardiac amyloidosis | |
|---|---|---|---|---|---|---|
| | 10% | 100% | 10% | 100% | 10% | 100% |
| *Supervised baseline* | | | | | | |
| Random Init. | $0.772 \pm 0.012$ | $0.831 \pm 0.004$ | $0.775 \pm 0.019$ | $0.868 \pm 0.003$ | $0.912 \pm 0.004$ | $0.945 \pm 0.001$ |
| *ECG-only SSL* | | | | | | |
| 3KG | $0.751 \pm 0.007$ | $0.823 \pm 0.007$ | $0.725 \pm 0.008$ | $0.853 \pm 0.015$ | $0.900 \pm 0.007$ | $0.945 \pm 0.002$ |
| CLOCS(CMSC) | $0.782 \pm 0.005$ | $0.815 \pm 0.009$ | $0.716 \pm 0.024$ | $0.857 \pm 0.013$ | $0.912 \pm 0.006$ | $0.948 \pm 0.001$ |
| PCLR | $0.812 \pm 0.004$ | $0.828 \pm 0.005$ | $0.722 \pm 0.032$ | $0.853 \pm 0.018$ | $0.924 \pm 0.003$ | $0.951 \pm 0.001$ |
| *Our models* | | | | | | |
| sEHR-ECG-Text | $\mathbf{0.880 \pm 0.003}$ | $0.888 \pm 0.005$ | $\mathbf{0.862 \pm 0.005}$ | $\mathbf{0.905 \pm 0.003}$ | $\mathbf{0.948 \pm 0.003}$ | $\mathbf{0.961 \pm 0.002}$ |
| ECG-sEHR | $\mathbf{0.881 \pm 0.003}$ | $\mathbf{0.892 \pm 0.001}$ | $\mathbf{0.866 \pm 0.008}$ | $\mathbf{0.902 \pm 0.003}$ | $\mathbf{0.949 \pm 0.001}$ | $\mathbf{0.961 \pm 0.002}$ |
| ECG-Text | $0.871 \pm 0.002$ | $0.880 \pm 0.001$ | $\mathbf{0.857 \pm 0.008}$ | $0.892 \pm 0.004$ | $0.940 \pm 0.002$ | $\mathbf{0.959 \pm 0.001}$ |

Table 3: Results (AUROC, mean and standard deviation over 5 runs with different random seeds) for pulmonary hypertension, low LVEF, and AFib in NSR classification tasks: (a) linear classification, (b) fine-tuned classification. Results within 95% confidence intervals of the best result are shown in **bold**. See Appendix D for AUPRC metrics.

(a) Linear classification

| | Pulmonary hypertension | | | Low LVEF | | | AFib in NSR | | |
|---|---|---|---|---|---|---|---|---|---|
| Method | 1% | 10% | 100% | 1% | 10% | 100% | 1% | 10% | 100% |
| *Supervised baseline* | | | | | | | | | |
| Random Init. | 0.805 ± 0.008 | 0.887 ± 0.005 | 0.927 ± 0.002 | 0.857 ± 0.006 | 0.919 ± 0.002 | 0.944 ± 0.000 | 0.834 ± 0.013 | 0.896 ± 0.002 | 0.922 ± 0.001 |
| *ECG-only SSL* | | | | | | | | | |
| 3KG | 0.779 ± 0.018 | 0.862 ± 0.002 | 0.874 ± 0.000 | 0.846 ± 0.016 | 0.902 ± 0.002 | 0.914 ± 0.000 | 0.824 ± 0.009 | 0.866 ± 0.000 | 0.870 ± 0.000 |
| CLOCS(CMSC) | 0.782 ± 0.021 | 0.857 ± 0.002 | 0.872 ± 0.000 | 0.864 ± 0.011 | 0.905 ± 0.002 | 0.916 ± 0.000 | 0.818 ± 0.009 | 0.862 ± 0.000 | 0.868 ± 0.000 |
| PCLR | 0.845 ± 0.016 | 0.894 ± 0.001 | 0.907 ± 0.000 | 0.894 ± 0.006 | 0.927 ± 0.000 | 0.936 ± 0.000 | 0.865 ± 0.007 | 0.898 ± 0.000 | 0.904 ± 0.000 |
| *Our models* | | | | | | | | | |
| sEHR-ECG-Text | **0.900 ± 0.005** | **0.932 ± 0.001** | 0.939 ± 0.000 | **0.911 ± 0.013** | **0.941 ± 0.001** | **0.951 ± 0.000** | **0.902 ± 0.008** | 0.928 ± 0.001 | 0.931 ± 0.000 |
| ECG-sEHR | **0.899 ± 0.007** | **0.933 ± 0.001** | **0.940 ± 0.000** | 0.910 ± 0.014 | **0.942 ± 0.001** | **0.951 ± 0.000** | **0.902 ± 0.007** | **0.930 ± 0.001** | **0.933 ± 0.000** |
| ECG-Text | 0.889 ± 0.006 | 0.924 ± 0.001 | 0.931 ± 0.000 | **0.901 ± 0.014** | 0.938 ± 0.001 | 0.947 ± 0.000 | 0.891 ± 0.008 | 0.923 ± 0.000 | 0.925 ± 0.000 |

(b) Fine-tuned classification

| | Pulmonary hypertension | | | Low LVEF | | | AFib in NSR | | |
|---|---|---|---|---|---|---|---|---|---|
| Method | 1% | 10% | 100% | 1% | 10% | 100% | 1% | 10% | 100% |
| *Supervised baseline* | | | | | | | | | |
| Random Init. | 0.805 ± 0.008 | 0.887 ± 0.005 | 0.927 ± 0.002 | 0.857 ± 0.006 | 0.919 ± 0.002 | 0.944 ± 0.000 | 0.834 ± 0.013 | 0.896 ± 0.002 | 0.922 ± 0.001 |
| *ECG-only SSL* | | | | | | | | | |
| 3KG | 0.800 ± 0.010 | 0.891 ± 0.002 | 0.925 ± 0.001 | 0.868 ± 0.007 | 0.922 ± 0.002 | 0.945 ± 0.000 | 0.842 ± 0.009 | 0.900 ± 0.001 | 0.922 ± 0.001 |
| CLOCS(CMSC) | 0.833 ± 0.002 | 0.892 ± 0.003 | 0.927 ± 0.001 | 0.893 ± 0.002 | 0.926 ± 0.002 | 0.945 ± 0.001 | 0.849 ± 0.005 | 0.897 ± 0.002 | 0.920 ± 0.001 |
| PCLR | 0.865 ± 0.010 | 0.911 ± 0.002 | 0.933 ± 0.001 | 0.905 ± 0.002 | **0.937 ± 0.002** | **0.949 ± 0.000** | 0.874 ± 0.006 | 0.906 ± 0.002 | 0.922 ± 0.001 |
| *Our models* | | | | | | | | | |
| sEHR-ECG-Text | **0.916 ± 0.004** | **0.934 ± 0.001** | 0.938 ± 0.003 | **0.933 ± 0.002** | 0.939 ± 0.003 | 0.949 ± 0.001 | **0.909 ± 0.008** | 0.919 ± 0.002 | **0.926 ± 0.001** |
| ECG-sEHR | **0.917 ± 0.005** | **0.935 ± 0.001** | **0.940 ± 0.001** | **0.932 ± 0.001** | 0.939 ± 0.002 | 0.949 ± 0.000 | **0.914 ± 0.005** | 0.920 ± 0.003 | **0.928 ± 0.001** |
| ECG-Text | 0.907 ± 0.003 | 0.926 ± 0.001 | 0.934 ± 0.001 | 0.927 ± 0.004 | **0.938 ± 0.001** | 0.948 ± 0.001 | 0.901 ± 0.008 | 0.916 ± 0.003 | **0.926 ± 0.002** |

blocks, etc.). These diagnoses are well-documented in the textual modality, particularly within ECG reports. As a result, the ECG-Text model performs best on these external datasets. In contrast, our internal datasets contain diseases for which labels are derived from EHR data. These labels are better captured by the sEHR modality. Therefore, the ECG-sEHR model outperforms the ECG-Text model when applied to internal datasets. It's worth noting that the sEHR-ECG-Text model demonstrates strong generalization across all internal datasets and achieves comparable performance with the ECG-Text model in the case of external datasets, i.e., PhysioNet2020 and Chapman, as it is trained with both sEHR and text modalities. Furthermore, it offers the advantage of comparing different modalities for retrieval tasks.

### 4.5.2 Retrieval Tasks

Following (Zhang et al., 2022), we also evaluate the pre-trained models on two zero-shot retrieval tasks: (i) Zero-shot ECG-ECG Retrieval, and (ii) Zero-shot Text-ECG Retrieval. We used data only from the global test split to create the queries and candidates for the retrieval tasks. For a given query, we rank the candidates by computing the cosine similarity between the representations of the query and the candidates obtained from pre-trained encoders. For the Text-ECG retrieval task, we obtain ECG and text embeddings from shared ECG-Text embedding space $\Omega_{et}$. We report the precision@$k$ metric for $k$=100, 500, and 1000, which represents the percentage of top $k$ ranked candidates that are relevant to the query.

**Zero-shot ECG-ECG Retrieval.** We take 1000 different ECGs as search queries for each of the 41 cardiovascular conditions that are based on ECG reports. For every condition, we select 100,000 candidate ECGs, of which 10,000 are classified as positive for the condition and 90,000 are classified as negative for the condition. The query ECGs and positive candidate ECGs are completely exclusive.

**Zero-shot Text-ECG Retrieval.** We take 1000 distinct ECG reports as search queries for each of the 41 cardiovascular conditions. For each of the conditions, we take 100,000 ECGs, out of which 10,000 ECGs show the condition and 90,000 ECGs have no connection to the condition. We also make sure that no ECG corresponding to the 1000 distinct ECG report queries is chosen as a candidate for the positive set.

Table 4 shows the zero-shot ECG-ECG retrieval and ECG-Text retrieval results. Our ECG-Text model outperforms the random guess method and ECG-only contrastive learning models by a large margin on both tasks.

Table 4: Zero-shot ECG-ECG retrieval and Text-ECG retrieval results. The *random* category results are from random guesses. P@$k$ denotes precision@$k$.

| | ECG-ECG Retrieval | | | Text-ECG Retrieval | | |
|---|---|---|---|---|---|---|
| Method | P@100 | P@500 | P@1000 | P@100 | P@500 | P@1000 |
| Random | 10.00 | 10.00 | 10.00 | 10.00 | 10.00 | 10.00 |
| PCLR | 38.41 | 35.34 | 33.74 | - | - | - |
| 3KG | 40.35 | 37.34 | 35.74 | - | - | - |
| CLOCS | 41.13 | 37.63 | 35.82 | - | - | - |
| ECG-Text | **55.13** | **49.47** | **46.33** | **73.02** | **66.92** | **63.58** |

### 4.5.3 Out-of-Distribution Detection

It is observed that the representations learned via self-supervised learning techniques help to better distinguish between *in-distribution* (IND) and *out-of-distribution* (OOD) datasets. We demonstrate this using representations obtained from our ECG-sEHR model to differentiate between two disparate ECG datasets. We take the proprietary ECG pulmonary hypertension (PH) cohort as the IND dataset and Holter ECGs (ECG recorded continuously over 24 hours or longer) from the open-source St Petersburg INCART 12-lead Arrhythmia Database (Tihonenko et al., 2008) as the OOD dataset. Non-overlapping 10-second long segments are taken from Holter ECGs and resampled to 500Hz to be consistent

Table 5: Out-of-distribution detection results using ECG representations obtained from the PH disease model and generic ECG representations obtained from the ECG-sEHR model. *sig.* denotes significance level.

| Metric | ECG-sEHR | PH |
|---|---|---|
| Rejection at 1% sig. (%) | 13.94 | 10.35 |
| Rejection at 5% sig. (%) | 49.67 | 29.87 |
| IND vs. OOD AUROC | 0.757 | 0.620 |

with ECGs from the IND PH dataset. We use the *relative Mahalanobis distance* (RMD) (Ren et al., 2021) method which is based on the Mahalanobis distance of embeddings from the distribution of the nearest predicted class, to determine whether the data is in-distribution or out-of-distribution. We compare the results obtained using representations extracted from the ECG-encoder of the pre-trained ECG-sEHR model with that of the representations obtained from the penultimate layer of the PH binary classifier trained from scratch on the PH cohort and show that the rejection rate at different significance levels is much higher in the case of ECG-sEHR model. The results are given in Table 5, which clearly shows that generic ECG representations are better at detecting out-of-distribution data compared to disease-specific representations.

## 5 Conclusion

Our work introduces a series of three multi-modal contrastive learning models. These models leverage both structured and unstructured EHRs to produce high-quality ECG representations. We have demonstrated that our pre-trained models outperform randomly initialized models and other ECG-only contrastive learning models by a wide margin on classification and retrieval tasks. Specifically, we perform the classification tasks using ECGs on cardiovascular diseases whose definitive diagnoses are obtained from more expensive and/or invasive tests in clinical settings. This is a significant breakthrough as ECG tests are widely available, non-invasive, and less expensive. Furthermore, our ECG representations have been shown to excel in detecting out-of-distribution data when compared to disease-specific representations.

## 6 Future Work

In this work, we make use of both structured EHRs and unstructured EHR text data to learn ECG representations. However, there are additional modalities present in unstructured EHRs such as images (MRI scans, CT scans, X-rays, and histopathology images related to cardiology), videos (echocardiograms/heart ultrasounds), and time-series signals (heart and lung sounds), which can provide even more meaningful infor-

mation through multi-modal contrastive learning. Another interesting future work would be the integration of federated learning frameworks to leverage multi-institutional medical data. This approach aims to capture a more diverse range of patient information, leading to enhanced ECG representation learning. While the disease models presented in this study undergo training and testing using real-world datasets, it is of utmost importance to conduct clinical validation across a diverse set of health systems before deploying them. This ensures that the models are equitable, unbiased, and trustworthy. We hope our work will serve as an inspiration for future endeavors in harnessing multi-modal EHR data to learn robust ECG representations.

## 7 Acknowledgements

We would like to thank Dr. Rickey E. Carter for insightful discussion and valuable feedback on this study.

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

# Appendix

## A  Dataset Details

### A.1  Pre-training Dataset Details

Table 6: The number of ECGs and the number of patients used during pre-training of the proposed models, i.e., sEHR-BERT, sEHR-ECG-Text, ECG-sEHR, and ECG-Text.

| Model | Train #ECGs | Train #Patients | Validation #ECGs | Validation #Patients | Test #ECGs | Test #Patients | Total #ECGs | Total #Patients |
|---|---|---|---|---|---|---|---|---|
| Global Splits | 5,479,435 | 1,463,009 | 450,775 | 121,932 | 3,210,110 | 853,477 | 9,140,320 | 2,438,418 |
| sEHR-BERT | - | 1,167,991 | - | 97,333 | - | - | - | - |
| sEHR-ECG-Text | 4,526,686 | 1,177,903 | 371,367 | 98,013 | - | - | - | - |
| ECG-sEHR | 4,553,278 | 1,196,478 | 373,649 | 99,608 | - | - | - | - |
| ECG-Text | 5,416,467 | 1,423,999 | 445,342 | 118,628 | - | - | - | - |

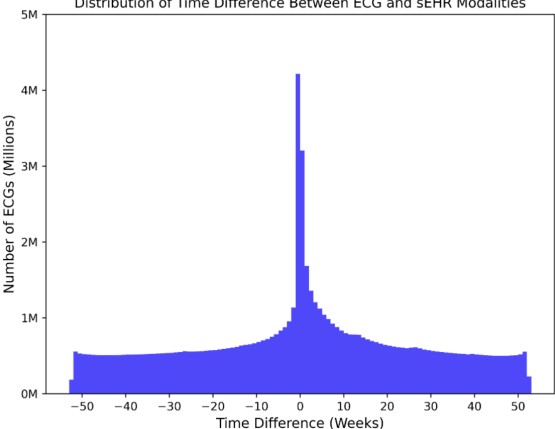
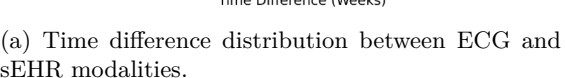

(a) Time difference distribution between ECG and sEHR modalities.

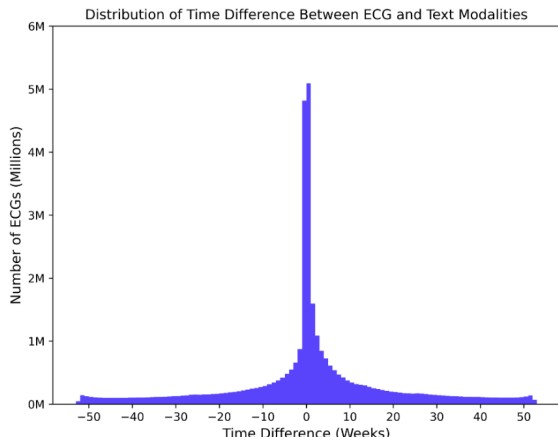

(b) Time difference distribution between ECG and text modalities.

Figure 3: The figures (a) and (b) show the distribution of time difference between ECG and other modalities: (a) the distribution of time difference between ECG and sEHR modalities, (b) the distribution of time difference between ECG and text modalities.

Table 6 outlines the number of ECGs and the number of patients used during pre-training of the proposed models. Figure 3a shows the time difference distribution between ECG and sEHR modalities, while Figure 3b displays the time difference distribution between ECG and text modalities. The Y-axis in Figure 3a represents the number of ECGs associated with at least one medical code for the sEHR modality in one-week intervals and the Y-axis in Figure 3b represents the number of ECGs associated with at least one biomedical entity (mentioned in Section 4.2.3) for the text modality in one-week intervals. The distributions are plotted using one-year time window (i.e., approximately 52 weeks) around ECG acquisition timestamp.

## A.2 Internal Classification Dataset Details

Table 7: Summary of the disease labeling process for coronary atherosclerosis, myocarditis, cardiac amyloidosis, pulmonary hypertension, and low LVEF. The *Case* column details the identification of patients with the specific disease, the *Control* column explains how control patients are produced for the corresponding disease, and the *Time window* indicates the time frame for associating ECGs with diagnoses. $(d_1, d_2)$ denotes the time frame extending $d_1$ days before and $d_2$ days after the first diagnosis timestamp. All ECGs of the control patients are taken into the control cohort. The abbreviations CAC, TTE, mPAP, TRV, and LVEF represent coronary artery calcium, transthoracic echocardiogram, mean pulmonary arterial pressure, tricuspid regurgitation velocity, and left ventricular ejection fraction, respectively.

| Disease | Case | Control | Time window |
|---|---|---|---|
| Coronary atherosclerosis | CAC score $\geq$ 300 | CAC score = 0 | (365, 365) |
| Myocarditis | Manual curation by expert physicians using EHR data | No history of myocarditis | (7, 7) |
| Cardiac amyloidosis | Manual curation by expert physicians using EHR data | Patients who have undergone TTE and no history of amyloidosis | (180, 180) |
| Pulmonary hypertension | mPAP $\geq$ 25 mmHg or TRV $\geq$ 3.4 m/s | mPAP $\leq$ 20 mmHg or TRV $\leq$ 2.8 m/s | (30, 30) |
| Low LVEF | LVEF $\leq$ 40 | LVEF > 40 | (14, 14) |

Table 8: Number of ECGs, number of patients, and disease prevalence in coronary atherosclerosis, myocarditis, cardiac amyloidosis, pulmonary hypertension, low LVEF, and AFib in NSR classification datasets.

| | Train | | | Validation | | | Test | | |
|---|---|---|---|---|---|---|---|---|---|
| Disease | #ECGs | #Patients | Prev.(%) | #ECGs | #Patients | Prev.(%) | #ECGs | #Patients | Prev.(%) |
| Coronary atherosclerosis | 19,281 | 10,589 | 38.78 | 1,604 | 870 | 39.66 | 11,290 | 6,088 | 39.13 |
| Myocarditis | 53,432 | 52,299 | 0.97 | 4,366 | 4,260 | 1.17 | 31,715 | 30,984 | 1.02 |
| Cardiac amyloidosis | 34,465 | 20,011 | 8.54 | 2,795 | 2,462 | 8.04 | 20,071 | 17,508 | 8.51 |
| Pulmonary hypertension | 200,777 | 73,908 | 14.71 | 16,132 | 6,091 | 14.10 | 115,602 | 42,893 | 14.34 |
| Low LVEF | 166,702 | 166,702 | 7.58 | 13,814 | 13,814 | 7.69 | 97,109 | 97,109 | 7.48 |
| AFib in NSR | 1,455,626 | 514,871 | 7.28 | 42,627 | 42,627 | 7.45 | 301,022 | 301,022 | 7.32 |

In this section, we summarize the disease labeling process and outline the number of ECGs, the number of patients, and the disease prevalence used during training, validation, and testing of internal disease classification models. Table 7 shows the summary of the disease labeling process for coronary atherosclerosis, myocarditis, cardiac amyloidosis, pulmonary hypertension, and low LVEF. In the case of AFib in NSR, for the case-cohort, we encompass all NSR ECGs: those from 30 days before the first occurrence of AFib, during the occurrences, and extending 30 days after the last occurrence of AFib. The control cohort comprises all NSR ECGs obtained from patients with no evidence of AFib. Table 8 shows the number of ECGs, the number of patients, and the prevalence for each disease.

## A.3 External Classification Dataset Details

In this section, we provide an overview of the number of ECGs and the number of patients in PhysioNet2020 and Chapman datasets (see Table 9). The physioNet2020 dataset consists of ECG signals from multiple sources with varying signal lengths and sampling rates.

Table 9: Details of the PhysioNet2020 and Chapman datasets, including the number of ECGs, number of patients, signal lengths, and sampling rates.

| Dataset | #ECGs | #Patients | Signal length | Sampling rate |
|---|---|---|---|---|
| **PhysioNet2020** | | | | |
| CPSC2018 | 6,877 | 6,877 | 6-60 secs | 500 Hz |
| CPSC extra | 3,453 | 3,453 | 6-60 secs | 500 Hz |
| St Petersburg INCART | 74 | 32 | 30 mins | 257 Hz |
| PTB | 516 | 516 | - | 1000Hz |
| PTB-XL | 21,837 | 21,837 | 10 secs | 500 Hz |
| Georgia | 10,344 | 10,344 | 10 secs | 500 Hz |
| **Chapman** | | | | |
| Chapman | 10,646 | 10,646 | 10 secs | 500 Hz |

## B Model Architectures and Implementation Details

### B.1 Training Details

We execute all pre-training and classification tasks using 2 Nvidia V100 (16G) GPUs. However, for the pretraining tasks involving the text domain, we utilize 2 Nvidia A100 (40G) GPUs. All the original ECGs consist of 12 leads and are 10 seconds long with a sampling rate of 500Hz. During training, we use a random crop of 5 seconds in length, i.e., 2500 samples. We optimize all pre-training and classification models with AdamW optimizer (Loshchilov & Hutter, 2019) with $(\beta_1, \beta_2)$ set to (0.9, 0.999).

**Pre-training Details** We initially pre-train sEHR-BERT as described in Section 3.1. For multi-modal contrastive pre-training, we initialize the sEHR encoder with sEHR-BERT model weights and the text encoder with GatorTron-base (Yang et al., 2022) model weights. GatorTron-base (Yang et al., 2022) is a 345M-parameter language model pre-trained on large amounts of de-identified clinical notes (80B words) from the University of Florida Health System, having a vocabulary of size 50176. We use BERT and Megatron-BERT implementation offered by the Huggingface transformers library (Wolf et al., 2020) for sEHR and text encoders respectively. For the ECG encoder, we use ResNet-like architecture (He et al., 2016) customized to 1 dimension which consists of around 1M parameters. The ECG encoder is initialized randomly. In joint pre-training, we freeze the first 3 and 18 layers of sEHR and text encoders respectively, and fine-tune the remaining layers. Following Zhang et al. (2022), we set $\tau$ to 0.1. We assign equal weighting to both the directions of contrastive learning, i.e., from ECG to sEHR and sEHR to ECG, similarly for ECG and text, i.e., $(\lambda_{es}, \lambda_{et})$ is set to (0.5, 0.5). We used a batch size of 256 and an initial learning rate of 1e-4 for our models. For ECG-only contrastive learning models, we used a batch size of 512 and an initial learning rate of 1e-3. The learning rate is reduced by a factor of 2 if the validation loss stops decreasing continuously for 2 epochs and we early stop the training based on validation loss with an early stopping patience of 10 epochs.

**Classification Details** For classification tasks, we add a two-layered MLP head on top of the ECG encoder. We also add dropout layers after each hidden layer with a dropout probability of 0.2 for regularisation. A batch size of 128 is used for all classification models. We used an initial learning rate of 1e-3 for random initialization training for all diseases. For fine-tuning, we used an initial learning rate of 1e-3 for coronary atherosclerosis and myocarditis tasks, and 1e-4 for cardiac amyloidosis, pulmonary hypertension, low LVEF, and AFib in NSR tasks. The learning rate is reduced by a factor of 2 if the validation score stops increasing continuously for 2 epochs and we early stop the training based on validation loss with an early stopping patience of 10 epochs. During fine-tuning, we initialize the ECG encoder with the pre-trained ECG encoder weights and warm up the classification head (MLP) for 1024 steps by freezing the backbone network weights and then fine-tuning the entire network. During prediction, we take 6 consecutive 5-second long crops with

a stride of 1 second from the original 10-second long ECG. The median of the predictions of these 6 crops is taken as the final prediction for computing the AUROC score.

## B.2 Multi-Modal Architectures

In this section, we present the precise architectures of various multi-modal contrastive learning models. Figure 4a and Figure 4b show the architectures of the ECG-sEHR model, described in Section 3.3, and the ECG-Text model described in Section 3.4, respectively. Figure 4c shows the architecture of sEHR-ECG-Text shared space model mentioned in Section 3.2.

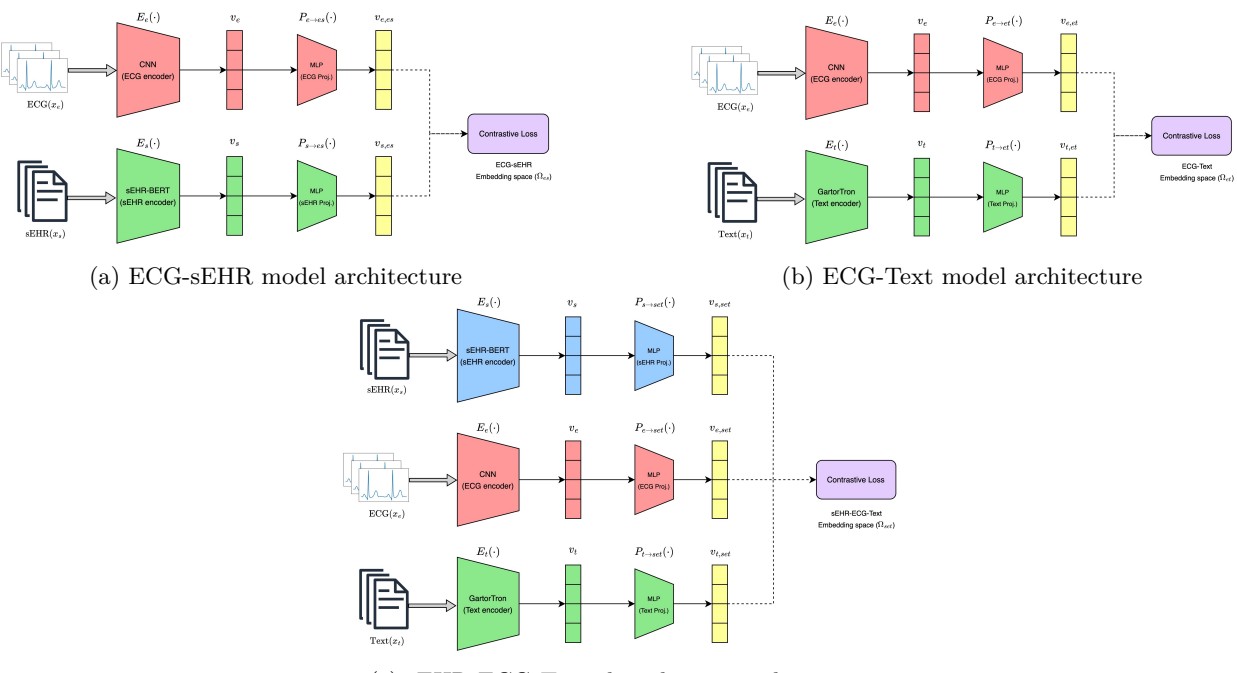

(a) ECG-sEHR model architecture

(b) ECG-Text model architecture

(c) sEHR-ECG-Text shared space architecture

Figure 4: The figures (a) and (b) show the architectures of bi-modal contrastive learning methods. (a) illustrates the architecture for the ECG-sEHR model, while (b) presents the architecture for the ECG-Text model. Figure (c) shows the shared space architecture of the sEHR-ECG-Text model, where contrastive loss between ECG and sEHR and, between ECG and text, is applied in sEHR-ECG-Text shared space ($\Omega_{set}$).

## B.3 ECG Encoder Architecture

We use ResNet-like architecture (He et al., 2016) customized to 1 dimension for time series ECG signals. This consisted of eight 1D convolution layers based on the basic block of ResNet. Details of each layer are given in Table 10. All convolutional layers employ batch normalization and ReLU activation function and a stride of 2. We add two fully connected layers with hidden sizes 128 and 64 on top of the backbone CNN architecture for classification tasks. We use the same architecture for all pre-training methods.

Table 10: ECG encoder architecture used for all experiments. IC, OC, and K represent the number of input channels, the number of output channels, and kernel size, respectively.

| Layer | Layer type | IC | OC | K |
|-------|-----------|-----|-----|---|
| 1 | Conv | 12 | 32 | 5 |
| 2 | Conv | 32 | 32 | 5 |
| 3 | Conv | 32 | 64 | 5 |
| 4 | Conv | 64 | 64 | 3 |
| 5 | Conv | 64 | 128 | 3 |
| 6 | Conv | 128 | 128 | 3 |
| 7 | Conv | 128 | 256 | 3 |
| 8 | Conv | 256 | 256 | 3 |

## C    Statistical Testing

We assess the statistical significance of our generalized model, sEHR-ECG-Text, by comparing it to each baseline using a two-sided $t$-test. This evaluation is conducted on linear classification tasks with 10% training splits. Each experiment is repeated 10 times, with 10 different fractional splits derived from the original training data. Our model significantly outperformed all the baselines with the $p$-value less than 1e-5. We show the average AUROC score over 10 runs along with 95% confidence intervals in Table 11.

Table 11: The table shows the average AUROC, with 95% confidence intervals in parentheses on linear classification tasks, i.e., coronary atherosclerosis, myocarditis, cardiac amyloidosis, pulmonary hypertension, low LVEF, and AFib in NSR, using 10% training splits. The results were calculated by conducting each experiment 10 times, using 10 different fractional splits obtained from the original training split. The sEHR-ECG-Text model significantly outperforms all the baselines with the $p$-value less than 1e-5.

| Method | Coronary atherosclerosis | Myocarditis | Cardiac amyloidosis | Pulmonary hypertension | Low LVEF | AFib in NSR |
|--------|--------------------------|-------------|---------------------|------------------------|----------|-------------|
| *Supervised baseline* | | | | | | |
| Random Init. | 0.770 (0.760, 0.779) | 0.778 (0.767, 0.789) | 0.910 (0.907, 0.913) | 0.886 (0.882, 0.890) | 0.917 (0.915, 0.920) | 0.897 (0.895, 0.898) |
| *ECG-only SSL* | | | | | | |
| 3KG | 0.721 (0.715, 0.727) | 0.686 (0.669, 0.704) | 0.869 (0.862, 0.875) | 0.862 (0.860, 0.863) | 0.902 (0.900, 0.903) | 0.866 (0.865, 0.866) |
| CLOCS(CMSC) | 0.736 (0.729, 0.742) | 0.660 (0.649, 0.671) | 0.879 (0.875, 0.883) | 0.857 (0.856, 0.859) | 0.905 (0.904, 0.906) | 0.861 (0.861, 0.862) |
| PCLR | 0.759 (0.757, 0.762) | 0.716 (0.704, 0.728) | 0.898 (0.894, 0.903) | 0.895 (0.894, 0.895) | 0.927 (0.927, 0.928) | 0.898 (0.897, 0.899) |
| *Our model* | | | | | | |
| sEHR-ECG-Text | 0.840 (0.838, 0.843) | 0.859 (0.846, 0.872) | 0.934 (0.930, 0.937) | 0.932 (0.931, 0.933) | 0.942 (0.941, 0.943) | 0.928 (0.928, 0.929) |

## D    Additional Evaluation Metrics

In this section, we provide the AUPRC scores for classification tasks performed on internal datasets. Table 12 shows AUPRC scores for coronary atherosclerosis, myocarditis, and cardiac amyloidosis classification tasks (an extension of Table 2) and Table 13 shows AUPRC scores for pulmonary hypertension, low LVEF, and AFib in NSR classification tasks (an extension of Table 3). We observe a similar trend in the AUPRC metric when comparing our models to baseline models, as we did with AUROC.

Table 12: An extension of Table 2 to include AUPRC scores (mean and standard deviation over 5 runs with different random seeds) for coronary atherosclerosis, myocarditis, and cardiac amyloidosis classification tasks: (a) linear classification, (b) fine-tuned classification. Results within 95% confidence intervals of the best result are shown in **bold**.

(a) Linear classification

| Method | Coronary atherosclerosis | | Myocarditis | | Cardiac amyloidosis | |
|---|---|---|---|---|---|---|
| | 10% | 100% | 10% | 100% | 10% | 100% |
| *Supervised baseline* | | | | | | |
| Random Init. | $0.686 \pm 0.013$ | $0.768 \pm 0.006$ | $0.075 \pm 0.019$ | $0.239 \pm 0.013$ | $0.687 \pm 0.009$ | $0.795 \pm 0.004$ |
| *ECG-only SSL* | | | | | | |
| 3KG | $0.643 \pm 0.014$ | $0.722 \pm 0.000$ | $0.047 \pm 0.013$ | $0.078 \pm 0.000$ | $0.522 \pm 0.020$ | $0.628 \pm 0.000$ |
| CLOCS(CMSC) | $0.661 \pm 0.011$ | $0.734 \pm 0.000$ | $0.033 \pm 0.005$ | $0.069 \pm 0.000$ | $0.534 \pm 0.014$ | $0.670 \pm 0.000$ |
| PCLR | $0.668 \pm 0.008$ | $0.761 \pm 0.000$ | $0.038 \pm 0.010$ | $0.082 \pm 0.001$ | $0.618 \pm 0.028$ | $0.721 \pm 0.001$ |
| *Our models* | | | | | | |
| sEHR-ECG-Text | $\mathbf{0.780 \pm 0.008}$ | $0.841 \pm 0.000$ | $\mathbf{0.200 \pm 0.031}$ | $\mathbf{0.304 \pm 0.001}$ | $\mathbf{0.775 \pm 0.017}$ | $0.842 \pm 0.000$ |
| ECG-sEHR | $\mathbf{0.773 \pm 0.018}$ | $\mathbf{0.843 \pm 0.000}$ | $\mathbf{0.178 \pm 0.028}$ | $0.291 \pm 0.001$ | $\mathbf{0.777 \pm 0.019}$ | $\mathbf{0.845 \pm 0.000}$ |
| ECG-Text | $0.754 \pm 0.011$ | $0.819 \pm 0.000$ | $0.122 \pm 0.029$ | $0.210 \pm 0.001$ | $0.692 \pm 0.021$ | $0.778 \pm 0.001$ |

(b) Fine-tuned classification

| Method | Coronary atherosclerosis | | Myocarditis | | Cardiac amyloidosis | |
|---|---|---|---|---|---|---|
| | 10% | 100% | 10% | 100% | 10% | 100% |
| *Supervised baseline* | | | | | | |
| Random Init. | $0.686 \pm 0.013$ | $0.768 \pm 0.006$ | $0.075 \pm 0.019$ | $0.239 \pm 0.013$ | $0.687 \pm 0.009$ | $0.795 \pm 0.004$ |
| *ECG-only SSL* | | | | | | |
| 3KG | $0.674 \pm 0.012$ | $0.756 \pm 0.009$ | $0.041 \pm 0.006$ | $0.227 \pm 0.031$ | $0.667 \pm 0.020$ | $0.798 \pm 0.004$ |
| CLOCS(CMSC) | $0.707 \pm 0.008$ | $0.747 \pm 0.011$ | $0.053 \pm 0.018$ | $0.255 \pm 0.015$ | $0.693 \pm 0.033$ | $0.808 \pm 0.002$ |
| PCLR | $0.735 \pm 0.010$ | $0.763 \pm 0.005$ | $0.043 \pm 0.009$ | $0.257 \pm 0.026$ | $0.750 \pm 0.010$ | $0.821 \pm 0.003$ |
| *Our models* | | | | | | |
| sEHR-ECG-Text | $\mathbf{0.822 \pm 0.006}$ | $0.838 \pm 0.007$ | $0.248 \pm 0.012$ | $\mathbf{0.343 \pm 0.019}$ | $\mathbf{0.805 \pm 0.007}$ | $\mathbf{0.845 \pm 0.005}$ |
| ECG-sEHR | $\mathbf{0.827 \pm 0.004}$ | $\mathbf{0.845 \pm 0.004}$ | $\mathbf{0.270 \pm 0.012}$ | $\mathbf{0.349 \pm 0.013}$ | $\mathbf{0.812 \pm 0.011}$ | $\mathbf{0.848 \pm 0.006}$ |
| ECG-Text | $0.813 \pm 0.006$ | $0.828 \pm 0.002$ | $0.196 \pm 0.017$ | $0.302 \pm 0.008$ | $0.779 \pm 0.006$ | $0.831 \pm 0.003$ |

# E    Ablation Study

We perform three ablations: (i) short ICD-10 vs. full ICD-10 codes in the sEHR modality as described in 4.2.2 using the ECG-sEHR model, (ii) report concatenation vs. entity concatenation in text modality as described in 4.2.3 using the ECG-Text model, and (iii) shared space vs. fine and coarse spaces using the sEHR-ECG-Text model as described in Section 3.2. A vocabulary of size 28593 and 42355 is used for short ICD-10 and full ICD-10 codes respectively. We show the ablation study results on the linear classification task in Table 14. The difference in performance between short ICD-10 and full ICD-10 codes is very minimal which can be attributed to the point that full ICD codes provide little to no extra information. In the report concatenation vs. entity concatenation ablation, entity concatenation yielded better results for the majority of the diseases. We speculate that this is because entity concatenation captures better long-term dependencies than report concatenation as we incorporate information from one-year time window around ECGs. Note that the sEHR-ECG-Text model with FAC spaces architecture yielded slightly better results than the shared space architecture.

Table 13: An extension of Table 3 to include AUPRC scores (mean and standard deviation over 5 runs with different random seeds) for pulmonary hypertension, low LVEF, and AFib in NSR classification tasks: (a) linear classification, (b) fine-tuned classification. Results within 95% confidence intervals of the best result are shown in **bold**.

(a) Linear classification

| Method | Pulmonary hypertension | | | Low LVEF | | | AFib in NSR | | |
|---|---|---|---|---|---|---|---|---|---|
| | 1% | 10% | 100% | 1% | 10% | 100% | 1% | 10% | 100% |
| *Supervised baseline* | | | | | | | | | |
| Random Init. | $0.420 \pm 0.011$ | $0.585 \pm 0.011$ | $0.710 \pm 0.005$ | $0.354 \pm 0.013$ | $0.527 \pm 0.012$ | $0.659 \pm 0.003$ | $0.356 \pm 0.030$ | $0.545 \pm 0.011$ | $0.642 \pm 0.004$ |
| *ECG-only SSL* | | | | | | | | | |
| 3KG | $0.385 \pm 0.022$ | $0.512 \pm 0.005$ | $0.537 \pm 0.000$ | $0.322 \pm 0.032$ | $0.480 \pm 0.006$ | $0.525 \pm 0.000$ | $0.375 \pm 0.016$ | $0.454 \pm 0.003$ | $0.466 \pm 0.000$ |
| CLOCS(CMSC) | $0.394 \pm 0.029$ | $0.521 \pm 0.003$ | $0.548 \pm 0.000$ | $0.369 \pm 0.030$ | $0.503 \pm 0.008$ | $0.542 \pm 0.000$ | $0.373 \pm 0.013$ | $0.457 \pm 0.002$ | $0.472 \pm 0.000$ |
| PCLR | $0.487 \pm 0.043$ | $0.605 \pm 0.007$ | $0.633 \pm 0.000$ | $0.467 \pm 0.036$ | $0.600 \pm 0.003$ | $0.630 \pm 0.000$ | $0.443 \pm 0.022$ | $0.539 \pm 0.003$ | $0.556 \pm 0.000$ |
| *Our models* | | | | | | | | | |
| sEHR-ECG-Text | $\mathbf{0.618 \pm 0.012}$ | $0.722 \pm 0.003$ | $0.743 \pm 0.000$ | $\mathbf{0.529 \pm 0.046}$ | $\mathbf{0.664 \pm 0.003}$ | $\mathbf{0.695 \pm 0.000}$ | $\mathbf{0.598 \pm 0.009}$ | $0.662 \pm 0.001$ | $0.670 \pm 0.000$ |
| ECG-sEHR | $\mathbf{0.627 \pm 0.020}$ | $\mathbf{0.727 \pm 0.002}$ | $\mathbf{0.747 \pm 0.000}$ | $\mathbf{0.542 \pm 0.052}$ | $\mathbf{0.663 \pm 0.004}$ | $0.693 \pm 0.000$ | $\mathbf{0.606 \pm 0.014}$ | $\mathbf{0.672 \pm 0.002}$ | $\mathbf{0.679 \pm 0.000}$ |
| ECG-Text | $0.595 \pm 0.013$ | $0.689 \pm 0.002$ | $0.713 \pm 0.000$ | $\mathbf{0.493 \pm 0.030}$ | $0.634 \pm 0.003$ | $0.668 \pm 0.000$ | $0.561 \pm 0.015$ | $0.637 \pm 0.002$ | $0.646 \pm 0.000$ |

(b) Fine-tuned classification

| Method | Pulmonary hypertension | | | Low LVEF | | | AFib in NSR | | |
|---|---|---|---|---|---|---|---|---|---|
| | 1% | 10% | 100% | 1% | 10% | 100% | 1% | 10% | 100% |
| *Supervised baseline* | | | | | | | | | |
| Random Init. | $0.420 \pm 0.011$ | $0.585 \pm 0.011$ | $0.710 \pm 0.005$ | $0.354 \pm 0.013$ | $0.527 \pm 0.012$ | $0.659 \pm 0.003$ | $0.356 \pm 0.030$ | $0.545 \pm 0.011$ | $0.642 \pm 0.004$ |
| *ECG-only SSL* | | | | | | | | | |
| 3KG | $0.429 \pm 0.015$ | $0.600 \pm 0.008$ | $0.703 \pm 0.004$ | $0.422 \pm 0.013$ | $0.582 \pm 0.007$ | $0.673 \pm 0.003$ | $0.430 \pm 0.018$ | $0.569 \pm 0.004$ | $0.640 \pm 0.004$ |
| CLOCS(CMSC) | $0.497 \pm 0.004$ | $0.605 \pm 0.014$ | $0.714 \pm 0.004$ | $0.483 \pm 0.011$ | $0.602 \pm 0.005$ | $0.671 \pm 0.006$ | $0.449 \pm 0.008$ | $0.563 \pm 0.004$ | $0.638 \pm 0.003$ |
| PCLR | $0.545 \pm 0.021$ | $0.660 \pm 0.010$ | $0.726 \pm 0.004$ | $0.546 \pm 0.004$ | $0.649 \pm 0.002$ | $\mathbf{0.692 \pm 0.002}$ | $0.509 \pm 0.013$ | $0.585 \pm 0.006$ | $0.643 \pm 0.002$ |
| *Our models* | | | | | | | | | |
| sEHR-ECG-Text | $\mathbf{0.668 \pm 0.012}$ | $0.723 \pm 0.006$ | $0.746 \pm 0.009$ | $\mathbf{0.651 \pm 0.004}$ | $0.660 \pm 0.008$ | $\mathbf{0.692 \pm 0.008}$ | $0.626 \pm 0.021$ | $0.631 \pm 0.006$ | $0.658 \pm 0.010$ |
| ECG-sEHR | $\mathbf{0.678 \pm 0.014}$ | $\mathbf{0.729 \pm 0.004}$ | $\mathbf{0.752 \pm 0.004}$ | $0.643 \pm 0.003$ | $\mathbf{0.664 \pm 0.007}$ | $\mathbf{0.695 \pm 0.005}$ | $\mathbf{0.640 \pm 0.014}$ | $\mathbf{0.634 \pm 0.009}$ | $\mathbf{0.661 \pm 0.003}$ |
| ECG-Text | $0.639 \pm 0.009$ | $0.694 \pm 0.010$ | $0.731 \pm 0.004$ | $0.618 \pm 0.008$ | $0.638 \pm 0.006$ | $0.678 \pm 0.004$ | $0.595 \pm 0.016$ | $0.616 \pm 0.008$ | $0.655 \pm 0.008$ |

Table 14: Results of the ablation study. Comparison of linear classification performance (AUROC, mean and standard deviation over 5 runs with different random seeds) between short ICD codes and full ICD codes (ECG-sEHR), report concatenation and entity concatenation (ECG-Text), and shared space versus fine and coarse spaces (sEHR-ECG-Text). The best results of each ablation are shown in **bold**.

| Method | Coronary atherosclerosis | Myocarditis | Cardiac amyloidosis | Pulmonary hypertension | Low LVEF | AFib in NSR |
|---|---|---|---|---|---|---|
| **ECG-sEHR** | | | | | | |
| Short ICD-10 codes | $\mathbf{0.891 \pm 0.000}$ | $\mathbf{0.896 \pm 0.000}$ | $\mathbf{0.959 \pm 0.000}$ | $\mathbf{0.940 \pm 0.000}$ | $\mathbf{0.951 \pm 0.000}$ | $0.933 \pm 0.000$ |
| Full ICD-10 codes | $0.888 \pm 0.000$ | $0.894 \pm 0.000$ | $0.958 \pm 0.000$ | $\mathbf{0.940 \pm 0.000}$ | $\mathbf{0.951 \pm 0.000}$ | $\mathbf{0.934 \pm 0.000}$ |
| **ECG-Text** | | | | | | |
| Entity concatenation | $0.875 \pm 0.000$ | $0.881 \pm 0.000$ | $\mathbf{0.949 \pm 0.000}$ | $\mathbf{0.931 \pm 0.000}$ | $\mathbf{0.947 \pm 0.000}$ | $\mathbf{0.925 \pm 0.000}$ |
| Report concatenation | $\mathbf{0.879 \pm 0.000}$ | $\mathbf{0.895 \pm 0.001}$ | $0.939 \pm 0.000$ | $0.921 \pm 0.000$ | $0.938 \pm 0.000$ | $\mathbf{0.925 \pm 0.000}$ |
| **sEHR-ECG-Text** | | | | | | |
| Fine and coarse spaces | $\mathbf{0.890 \pm 0.000}$ | $\mathbf{0.901 \pm 0.001}$ | $\mathbf{0.960 \pm 0.000}$ | $\mathbf{0.939 \pm 0.000}$ | $\mathbf{0.951 \pm 0.000}$ | $\mathbf{0.931 \pm 0.000}$ |
| Shared space | $0.883 \pm 0.000$ | $0.884 \pm 0.001$ | $0.955 \pm 0.000$ | $0.936 \pm 0.000$ | $0.949 \pm 0.000$ | $0.928 \pm 0.000$ |

