# OpenReview forum: "ECG Representation Learning with Multi-Modal EHR Data"
_TMLR — Accepted by TMLR_

### Review · Reviewer_uNfp · 2023-09-26

**Summary Of Contributions:**

The paper is about representation learning for ECGs, and the authors consider how to perform multi-modal contrastive learning using a private dataset combining ECGs, structured EHRs (information about diagnoses, prescriptions, tests, etc) and unstructured EHRs (free-form text). It differs from previous work on self-supervised and contrastive learning as follows:

- SSL and contrastive learning are performed mainly in a uni-modal way for general domain images, or for multi-modal datasets combining text and images (e.g., captioned images scraped from the internet). Most applications of SSL/CL to medical data are also either uni-modal with images, or with text-image pairs. This work considers three modalities: ECG waveforms (time series), sEHRs (not quite text), and free-form text from unstructured EHRs.
- SSL and contrastive learning have been previously applied to ECGs but typically in a uni-modal fashion. One work combined ECGs with a different modality (labs and vitals time series), but the use of EHR data here seems novel.

The recipe for contrastive learning is usually straightforward, but the authors confront several difficulties here. One is determining the best encoders for each modality: they use a CNN for ECGs (a standard choice), a pre-trained LM for the unstructured EHR data (GatorTron), and a custom BERT-like model for the sEHR data. They must also consider how to compare between three modalities: as shown in Figure 1, there are separate embedding spaces in which two contrastive losses are calculated (ECG-sEHR, and ECG-sEHR-Text). This seems reasonable, and the loss functions (eqs. 1 and 2) are the standard losses for each space. Methodologically, this work is not too novel, but the choices are sensible and not straightforward to execute on (particularly training the custom BERT-like model for sEHR data).

The authors in fact train three separate multi-modal contrastive learning models, which utilize either type of EHR or both types. In addition, they train ECG-MTL, a multi-task model trained in a supervised manner for a large number of classification and regression tasks. (Actually, I could not understand whether this was a separate model, or fine-tuned from one of the multi-modal models - I request clarification on this below.)

The evaluations show that the new models are generally more effective than existing contrastive learning methods for a variety of downstream tasks, when tested with linear probing or full fine-tuning, and when tested on either the internal test set or external open-source datasets. Due to the strong existing backbones, the new models are also more sample-efficient.

The authors also show results for zero-shot retrieval tasks, and out-of-distribution detection, but while interesting, these seem less clinically relevant.

**Audience:**

Yes

**Broader Impact Concerns:**

I have no broader impact concerns.

**Claims And Evidence:**

Yes

**Requested Changes:**

A couple thoughts and questions:

- The results in Tables 2-4 are generally quite negative regarding the existing contrastive learning methods. They're rarely competitive with the new models (which is perhaps a good thing), but they're often worse than a simple supervised baseline. This is perhaps not a direct contradiction of previous work, as they did not use the same datasets, but it does suggest that those are ineffective representation learning methods. This is odd, because they use much more data and rely on sound contrastive learning principles. Can the authors comment on this, and perhaps make room in the paper to discuss why those methods don't work?

- The ImageNet comparison in contribution #3 was confusing to me. ImageNet is an open-source dataset, this is a model trained on a closed-source dataset. Can you rephrase this contribution so the comparison is more clear, or perhaps omit the comparison?

- As mentioned above, I could not tell whether ECG-MTL was trained in a purely supervised fashion or fine-tuned based on a backbone from the previous multi-modal models. If it's the former, I'm confused why it belongs in this paper. If it's the latter, I would have expected more discussion of this point in Section 3.5, and perhaps a comparison between backbones from which it's fine-tuned.

- In Section 3.2, the claim about the modalities having different levels of granularity is interesting, but this isn't accompanied by any empirical evidence or ablations. If the granularities were more similar, how would the learning approach differ? And could it be added as an ablation study? It's not that the results are strong as is, but from a scientific perspective, omitting any empirical evidence on this point is unsatisfying for interested readers.

- "GatorTron" is misspelled in Figure 1.

- Just a nit, but assuming L2 normalization when computing cosine similarity is unnecessary. The standard definition of cosine similarity performs the normalization internally ([PyTorch](https://pytorch.org/docs/stable/generated/torch.nn.CosineSimilarity.html), [Wikipedia](https://en.wikipedia.org/wiki/Cosine_similarity)).

- I couldn't tell why eq. 2 is repeated on the very next page in eq. 4.

- For context, can the authors provide some information about SOTA results on the external datasets? That information is missing, and it would be interesting to know whether this method enables better performance.

- The explanations about IRB approval don't seem to belong in the conclusion. Perhaps they should be moved earlier, for example to a section where the datasets are introduced?

**Strengths And Weaknesses:**

The main strengths of this work are formulating an approach to combine three separate data modalities, and showing convincing evidence that it leads to better representations than existing methods. I suspect this could encourage more work on combining diverse data modalities, which are likely available in other private datasets at medical institutions around the world. I generally appreciated the evaluations, particularly the careful splitting of data and the use of external datasets.

I don't see any important technical weaknesses. I ask a couple questions below, which may or may not merit some further experiments. As mentioned above, I don't find the approach very methodologically innovative, but it's reasonable and enables an interesting and challenging application of contrastive learning. One concern I would definitely like discussed is a clarification around ECG-MTL and why it's included in this paper if it's a purely supervised model.

---

> ### Author Response · Authors · 2023-10-28
> **Response to Reviewer uNfp (1/2)**
>
> Dear reviewer, thank you for the valuable feedback. We have incorporated your suggestions into our manuscript and have marked the changes in blue to facilitate review. We address your concerns in detail below.
>
> >The results in Tables 2-4 are generally quite negative regarding the existing contrastive learning methods. They're rarely competitive with the new models (which is perhaps a good thing), but they're often worse than a simple supervised baseline. This is perhaps not a direct contradiction of previous work, as they did not use the same datasets, but it does suggest that those are ineffective representation learning methods. This is odd, because they use much more data and rely on sound contrastive learning principles. Can the authors comment on this, and perhaps make room in the paper to discuss why those methods don't work?
>
> We hope the following evidence and explanation answers this question.
>
> **3KG and CLOCS correctness**
> * We updated the linear classification results on external datasets by following the specific guidelines outlined in 3KG [1] for the PhysioNet2020 dataset and CLOCS [2] for the Chapman dataset to replicate the state-of-the-art results presented in those works (see Section 4.3.2 in blue). We have also added results for supervised baseline trained with random initialization for comparison (Table 1).
> * Please note that 3KG performs well in comparison with the random initialization model, outperforming it on 1% splits and showing comparability on 10% splits for both PhysioNet2020 and Chapman datasets. Additionally, the results achieved with 3KG by us are on par with the results reported by 3KG authors (0.872 vs 0.890 on 100% split for PhysioNet2020 dataset). The results achieved with CLOCS by us are also on par with the results reported by CLOCS authors (0.946 vs. 0.959 on 100% split for Chapman dataset). We suspect these discrepancies arise from variations in the encoder and data distribution used for pre-training. 3KG employs the same dataset for both pre-training and evaluation, while we use our private dataset for pre-training. It's important to note that the PhysioNet2020 dataset contains Holter ECGs, which are out-of-distribution compared to our dataset. This evidence ensures the correctness of our 3KG and CLOCS implementation.
>
> **PCLR correctness**
> * We evaluated the PCLR [3] method using the [model](https://github.com/broadinstitute/ml4h/tree/master/model_zoo/PCLR) (trained on 3.2 million ECGs) released by PCLR authors on internal datasets and achieved comparable performance with our implementation of PCLR. The table below presents a comparison of linear classification performance (AUROC) between our PCLR model and the authors' released PCLR model. We suspect that the minor differences arise from variations in the ECG encoder. They use a larger model with 6.4M parameters, while our model consists of 1M parameters. This evidence ensures the correctness of our PCLR implementation.
>
> |       Disease          | Our PCLR model | Authors' released PCLR model |
> | :--------------------: | :----------------------: | :-----------------------------: |
> |Coronary Atherosclerosis |0.826|0.842|
> |Myocarditis        |  0.819| 0.817|
> |Cardiac Amyloidosis    | 0.929|0.936|
> | Pulmonary Hypertension   |0.908 |  0.903 |
> |Low LVEF| 0.936 |0.937|
> |AFib in NSR| 0.904|0.900 |
> * We suspect that the main reason for the poor performance of ECG-only contrastive learning models on internal datasets is their exclusive dependence on ECGs for learning representations, i.e., comparing different instances of ECG data. This may not be sufficient to learn the complex patterns of various medical conditions. In contrast, our method aligns ECGs with EHR data, providing rich contextual information about a patient's health history, including diagnoses, procedures, medications, and more. This approach is beneficial for learning ECG patterns more effectively. We discuss this in Section 4.5.1 (paragraph 1, point 5) in the revised document.
>
>
> [1] Gopal et al., 2021, 3kg: contrastive learning of 12-lead electrocardiograms using physiologically-inspired augmentations, ML4H, 2021
>
> [2] Kiyasseh et al., 2021, Clocs: Contrastive learning of cardiac signals across space, time, and patients, ICML, 2020
>
> [3] Diamant et al., 2022, Patient contrastive learning: A performant, expressive, and practical approach to electrocardiogram modeling, PLoS Computational Biology, 2022.
>
>
> >The ImageNet comparison in contribution #3 was confusing to me. ImageNet is an open-source dataset, this is a model trained on a closed-source dataset. Can you rephrase this contribution so the comparison is more clear, or perhaps omit the comparison?
>
> Thank you for bringing this to our attention. To avoid any confusion, we have chosen to omit the comparison in contribution #3.

---

> ### Author Response · Authors · 2023-10-28
> **Response to Reviewer uNfp (2/2)**
>
> >As mentioned above, I could not tell whether ECG-MTL was trained in a purely supervised fashion or fine-tuned based on a backbone from the previous multi-modal models. If it's the former, I'm confused why it belongs in this paper. If it's the latter, I would have expected more discussion of this point in Section 3.5, and perhaps a comparison between backbones from which it's fine-tuned.
>
> ECG-MTL is indeed a purely supervised model, trained from scratch with random weights initialization. After pre-training, we utilized features extracted from the backbone network (the layer before the MTL head) for linear evaluation, and the weights of backbone network for fine-tuning. However, as correctly pointed out, it appears that ECG-MTL may not be a suitable addition to this paper. In light of this, we have decided to remove the ECG-MTL model from our paper to maintain the paper's focus and clarity on multi-modal contrastive learning methods.
>
>
> >In Section 3.2, the claim about the modalities having different levels of granularity is interesting, but this isn't accompanied by any empirical evidence or ablations. If the granularities were more similar, how would the learning approach differ? And could it be added as an ablation study? It's not that the results are strong as is, but from a scientific perspective, omitting any empirical evidence on this point is unsatisfying for interested readers.
>
> If granularities were more similar, we could apply the contrastive objective between various modalities in single shared embedding space, an approach also highlighted in MultiModal Versatile Networks (MMV) [4]. This is done by projecting modality specific representations into joint sEHR-ECG-Text space $(\Omega_{set})$ using a single projection head $(P_{m \rightarrow set})$ for each input modality $m$. Our revised paper includes a comparative study of this shared space versus fine and coarse spaces (FAC) as part of the ablation analysis, with FAC being our main method. Please note that we had to move ablation study section to Appendix E for reasons of space. We also provide shared space architecture of sEHR-ECG-Text model in Appendix B.2.
>
> [4] Alayrac et al., 2020, Self-Supervised MultiModal Versatile Networks, NeurIPS, 2020
> >"GatorTron" is misspelled in Figure 1.
>
> Thank you for pointing out. We have corrected it in the revised paper.
>
>
> >Just a nit, but assuming L2 normalization when computing cosine similarity is unnecessary. The standard definition of cosine similarity performs the normalization internally (PyTorch, Wikipedia)
>
> We have revised the cosine similarity equation to conform to the standard definition.
>
>
> >I couldn't tell why eq. 2 is repeated on the very next page in eq. 4.
>
> * Equations 2 and 4 are different from each other. Equation 2 is utilized in the sEHR-ECG-Text model for contrastive learning between ECG and Text in the coarse-grained sEHR-ECG-Text embedding space $(\Omega_{set})$. This is used in conjunction with contrastive loss between ECG and sEHR, where loss between these two is applied in fine-grained ECG-sEHR embedding space $\Omega_{es}$) (Equation 1).
>
> * In contrast, Equation 4 is used in the ECG-Text model. Note that sEHR modality is not involved in this model, so there is no requirement for two embedding spaces. The contrastive loss between ECG and Text is applied in the joint ECG-Text space $(\Omega_{et})$. See that $z$ in $v_{m,z}$ (the representation of the input modality $x_m$ in the shared space $\Omega_{z}$) is $set$ in Equation 2 and $et$ in Equation 4. To provide more clarity, we have added this information in Section 3.4 in the revised paper. We also provide model architecture for ECG-sEHR and ECG-Text models in Appendix B.2, and have made references to it from the relevant sections in the main text where methodology is discussed.
>
>
> >For context, can the authors provide some information about SOTA results on the external datasets? That information is missing, and it would be interesting to know whether this method enables better performance.
>
> * On 100% training splits, the state-of-the-art (SOTA) linear classification performance (AUROC) on PhysioNet2020 is 0.890, as reported by 3KG, and 0.959 on Chapman, as reported by CLOCS. We outperform PhysioNet2020 by 2.5% (0.915 vs. 0.890) and Chapman by 3.1% (0.990 vs. 0.959).
> * It's worth noting that, on equal grounds, we surpass 3KG by 4.3% (0.915 vs 0.872) and CLOCS by 4.4% (0.990 vs. 0.946). We discuss the comparison of our models' performance with previous best results in Section 4.5.1 (refer to paragraph 1, point 4).
>
>
> >The explanations about IRB approval don't seem to belong in the conclusion. Perhaps they should be moved earlier, for example to a section where the datasets are introduced?
>
> Thank you for the suggestion. We have now moved explanations about IRB approval to Section 4.1.

---

### Review · Reviewer_323V · 2023-10-02

**Summary Of Contributions:**

The core value of this work lies in: 1) proposing the first instantiation of multi-modal contrastive learning framework for the purpose of learning representations of the ECG signal, 2) empirical evaluation of the proposed methods and ablating some of its domain-specific design choices.

**Audience:**

Yes

**Broader Impact Concerns:**

No special concerns beyond the standard caution about using ML in the health domain. The authors acknowledge those appropriately.

**Claims And Evidence:**

Yes

**Requested Changes:**

* In my opinion, must haves for acceptance: Given that both the training and the evaluation of the model is done using a closed-access, internal dataset, I think it particularly important for the paper to include as detailed empirical evaluation of the proposed methods as possible. Hence I would really like to see the following in the evaluation section: A) standard deviations of the results over multiple training trials (at least, over multiple fine-tuning trials with different random initializations of the linear head and random order of the data presented to the network; at best, multiple trials of SSL-based pre-training + finetuning) and bold the results for all the methods for which the standard deviations overlap considerably. B) Report the AUPRC metric as well, which seems important to me when reporting the classification performance on severely class-imbalanced data distribution.
* Extra: My understanding is that the multimodal-SSL-initialized models (for which the results are reported in the tables) are finetuned individually on each one of the classification tasks in Table 4. I would find it interesting to add the Multitask-Learning Component on top of the SSL-pretraining, as this would allow to assess the potential of combining the multi-modal SSL representation learning with multitask learning.

**Strengths And Weaknesses:**

* I find the paper quite well written, sufficiently clearly describing the methodology followed, and the motivation/reasoning behind it.
* I found the description of operationalization of the idea of multi-modal contrastive learning in this healthcare setting interesting, and I think it can benefit readers from this community.
* In my understanding, the paper seems technically correct. I did not find any errors in the operation of the method or the empirical evaluation protocol. That said, I am not very familiar with all of the types of data the authors work with, so it is possible I am overlooking something. The conclusions are supported by the empirical evidence presented.
* However, I think that the empirical evaluation of the methods should be slightly improved before acceptance (see below).
* The authors seem to characterize the novelty of the work appropriately, although I cannot warrant it's correct, because I am not sufficiently familiar with the literature on the use of contemporary representation learning methods in the medical/ECG-modality domain.

---

> ### Author Response · Authors · 2023-10-28
> **Response to Reviewer 323V**
>
> Dear reviewer, thank you for acknowledging our work and the valuable feedback. We have incorporated your suggestions into our manuscript and have marked the changes in blue to facilitate review. We provide the responses to the comments below.
>
>
> >In my opinion, must haves for acceptance: Given that both the training and the evaluation of the model is done using a closed-access, internal dataset, I think it particularly important for the paper to include as detailed empirical evaluation of the proposed methods as possible. Hence I would really like to see the following in the evaluation section: A) standard deviations of the results over multiple training trials (at least, over multiple fine-tuning trials with different random initializations of the linear head and random order of the data presented to the network; at best, multiple trials of SSL-based pre-training + finetuning) and bold the results for all the methods for which the standard deviations overlap considerably. B) Report the AUPRC metric as well, which seems important to me when reporting the classification performance on severely class-imbalanced data distribution.
>
>
> * We have included standard deviations of the results over multiple training trials for both linear classification and fine-tuning. Specifically, we reported the average and standard deviation over 5 trials with different random seeds. On 1% and 10% splits, for each experiment, we employed 5 different fractional splits derived from the original 100% training split and used different random seeds for each fractional split.
>
> * As per your suggestion to bold the results for all the methods for which the standard deviations overlap considerably, we showed the results within 1 standard deviation of the best result in bold.
>
> * Please see Table 1 for linear classification results (AUROC) on external datasets, Table 2 for classification results on coronary atherosclerosis, myocarditis, and cardiac amyloidosis diseases, and Table 3 for classification results on pulmonary hypertension, low LVEF, and AFib in NSR diseases in the revised document.
>
>
>
>
> * We have included the AUPRC metric in our evaluation as well. Please see Appendix D for the AUPRC performance in the revised paper.
>
> >Extra: My understanding is that the multimodal-SSL-initialized models (for which the results are reported in the tables) are finetuned individually on each one of the classification tasks in Table 4. I would find it interesting to add the Multitask-Learning Component on top of the SSL-pretraining, as this would allow to assess the potential of combining the multi-modal SSL representation learning with multitask learning.
>
>
> Your understanding is correct on fine-tuning with multimodal-SSL-initialized models on each one of the classification tasks. We highly appreciate your idea regarding the addition of a multitask-learning component on top of the SSL-pretraining. However, the number of ECGs within each disease dataset varies. Notably, the datasets for coronary atherosclerosis, myocarditis, and cardiac amyloidosis contain a relatively small number of ECGs (less than 50K), whereas the AFib in NSR dataset consists of a substantial 1.4 million ECGs. This discrepancy would lead to a significant number of missing labels when these datasets are merged. Therefore, it is unsuitable for effective multi-task learning.

---

> > ### Comment · Reviewer_323V · 2023-11-06
> >
> > Thank you for making the requested changes! I think they strengthen the paper.
> >
> > One more request: in the results tables 1-3 & 12-14, I would love to see the numbers bolded whenever the confidence intervals with the best result overlap - I've noticed several instances where that is not the case - I believe fixing this would allow those tables to tell a more consistent "story" (i.e. the conclusions would be more consistent between the columns).

---

> > > ### Author Response · Authors · 2023-11-08
> > > **Response to Reviewer 323V**
> > >
> > > Dear reviewer, thank you for acknowledging the updates.
> > >
> > > In response to your additional request, we have now bolded all the results whose mean AUROC overlaps with the 95% confidence intervals of the best result for each disease.

---

### Review · Reviewer_V79p · 2023-10-16

**Summary Of Contributions:**

In this paper, the authors propose a contrastive learning framework for downstream classification of ECG data. The framework includes two main components that learn similarities between ECG and structured EHR, and then ECG and textual data.

**Dataset:** The structured EHR data includes ICD codes (diagnosis/procedure codes) and medication prescriptions. The unstructured data contains ECG reports, ECHO reports, pathology reports, radiology reports, microbiology reports, clinical notes and surgical notes. The authors used an internal dataset and an external dataset.

**Evaluation:** The pre-training strategy is evaluated on various downstream tasks, such as classification, zero-shot retrieval, and out-of-distribution detection involving ECG as input only. They mainly use AUROC as a performance metric.

**Audience:**

Yes

**Broader Impact Concerns:**

The authors already discuss that the data is private and de-identified according to IRB standards.

**Claims And Evidence:**

No

**Requested Changes:**

Most of my comments are summarized above. One major request would be to make the code publicly available. Even if it's a proprietary dataset, the authors can provide the code for how they implement the losses and the training procedure, which would make it easier to follow the exact training steps described in the paper.

As it stands now, the paper is not ready for publication.

**Strengths And Weaknesses:**

**Strengths:**
- The proprietary dataset is very large in size.
- The model utilizes novel modalities for ECG representation learning via contrastive learning. This is quite creative as those modalities are not commonly used together.
- The work adopts the MultiModal Versatile Networks framework from Alayrac et al. 2020. Although it lacks methodological novelty, it can be considered a novel application of existing work.

**Weaknesses:**
- The proprietary dataset is private.
- The code is not shared, I would highly recommend that they share their repository considering that work in this area is generally based on private models.
- The exact contributions of the work are unclear as currently listed in the abstract. There are many results in the main manuscript, however it is also unclear which framework leads to the best improvements (ECG-sEHR vs ECG-Text vs ECG-sEHR-Text vs ECG-MTL). It currently seems that all models perform better in different scenarios, which does not really clarify the strengths of the proposed approach.
- It's also unclear to me the difference between ECG-sEHR-Text and ECG-MTL. Does it just take the ECG encoder from ECG-sEHR-Text and apply the MTL head? Do the other models perform single task classification compared to ECG-MTL?
- It is unclear why they conduct those specific ablation studies. It would be interesting to explore the impact of using different modalities, rather than long/short ICD and report/entity concatenation as those are not the main contributions of the work.
- Have the authors considered how this pre-training framework could generalize to the other input modalities (not just ECG)?
- There is no evidence of any hyper parameter tuning. This can help clarify the robustness of the proposed approaches with respect to one another.
- There is no statistical significance testing or confidence intervals provided. They also only report the average AUROC without any standard deviation.
- Have you considered reporting other relevant metrics? such as AUPRC
- The notation in the methodology section is confusing. It needs to be simplified and match the notation used in the main figure. They introduce general modality-agnostic notation, and then customize it to each modality. I would suggest sticking to one.
- The caption of the main figure should also explain all relevant notation without having to refer back to the main text.
- The figure caption also mentions that in both ECG-sEHR and ECG-Text pre-training, the projection mapping P(es->set) is not used. Isn't it used in the latter? Where is the ECG-Text embedding space in the figure?
- What's the motivation behind using the weighting scalars in the loss in equations 1 and 2?
- How is the loss in equation 3 used? From my understanding, you perform ECG-sEHR pertaining (section 3.3), then ECG-Text pertaining (section 3.4), then ECG-MTL? Please clarify.
- What's the difference between equation 4 and equation 2?
- Can you provide a figure that clarifies the final downstream architectures for all models, including ECG-MTL?
- Can you also provide a figure that clarifies how the modalities were linked with one another considering the varying timeframes / time differences between them?
- Can you provide a figure that clarifies the difference between report and entity concatenation? It is unclear and the section is poorly written.
- There are many decisions that seem to be made based on heuristics, such as keeping ICD codes that are associated with at least 50 patients.
- Section 4.3.1 describes how the samples were labeled, can you create a table that summarizes the labeling procedure of each disease label. The text is difficult to follow.
- The baselines are also not strong. Are there other multi-modal baselines that you can compare to? Did you perform any hyper parameter tuning for the baselines? What were the learning rates, etc.

**Presentation of the paper also needs to be significantly improved:**
- There are typos and grammar mistakes.
- The paper is quite dense and extensive and hence the authors need to clarify certain elements of the text as described below. It mostly reads as a technical report, such that the methodological contributions / design elements are integrated with implementation details (hyper parameters, etc.)
- It's uncommon to refer to EHR as EHRs
- The first paragraph in the introduction is too long and it's unclear what the main message is. This also applies to other sections in the text.
- "weekly" manner should be rephrased
- consistency of capitalization in introducing abbreviations
- The tables are referenced in random order. They also reference tables in the appendix in the main text without mentioning that they are supplementary.
- All table and figure captions must be elaborated extensively.
- Why is the AUROC reported as a percentage and to two decimal places? It is typically reported as a three decimal figure.

---

> ### Author Response · Authors · 2023-10-28
> **Response to Reviewer V79p (1/5)**
>
> Dear reviewer, we greatly appreciate your thorough review. We have incorporated your suggestions into our manuscript and highlighted the changes in blue. Below, we provide a comprehensive explanation to each comment in the following order to ensure flow and clarity.
> > It's also unclear to me the difference between ECG-sEHR-Text and ECG-MTL. Does it just take the ECG encoder from ECG-sEHR-Text and apply the MTL head? Do the other models perform single task classification compared to ECG-MTL?
>
> * No, ECG-MTL model doesn't take the ECG encoder from ECG-sEHR-Text. ECG-MTL is a supervised model trained with random weights initialization on 81 classification and 37 regression tasks. After pre-training, we utilized features extracted from the backbone network (the layer before the MTL head) for linear evaluation, and the weights of backbone network for fine-tuning. However, as raised by one of the other reviewers, it appears that ECG-MTL may not be a suitable addition to this paper as it is a purely supervised model. In light of this, we have decided to remove the ECG-MTL model to maintain the paper's focus and clarity on multi-modal contrastive learning methods.
>
> >The notation in the methodology section is confusing. It needs to be simplified and match the notation used in the main figure. They introduce general modality-agnostic notation, and then customize it to each modality. I would suggest sticking to one.
>
> >The caption of the main figure should also explain all relevant notation without having to refer back to the main text.
>
> * We apologize for the confusion. We adhered to modality-agnostic notation and matched the notations used in the main figure. We have also reorganized Section 3.2 for better flow and clarity of the methodology.
> * We have also explained all the notations in the caption of main figure.
>
> >How is the loss in equation 3 used? From my understanding, you perform ECG-sEHR pertaining (section 3.3), then ECG-Text pertaining (section 3.4), then ECG-MTL? Please clarify.
>
> * Please note that Equation 3, $\mathcal{L} = \mathcal{L_{es}} + \mathcal{L_{et}}$, represents the loss function for our main proposed model, **sEHR-ECG-Text**, described in Section 3.2. In this model, ECG-sEHR and ECG-Text matching are performed simultaneously.
> * The loss function incorporates two terms:
> 	* The first term, $\mathcal{L_{es}}$ (Equation 1), pertains to the contrastive loss between the ECG and sEHR modalities, which occurs in the fine-grained ECG-sEHR space $(\Omega_{es})$.
> 	* The second term, $\mathcal{L_{et}}$ (Equation 2), pertains to the contrastive loss between the ECG and text modalities, which occurs in the coarse-grained sEHR-ECG-Text space $(\Omega_{set})$.
> * These two loss terms are added together in Equation 3, and backpropagation is applied to optimize the model. ECG-MTL model doesn't come into play in this model.
>
> > The figure caption also mentions that in both ECG-sEHR and ECG-Text pre-training, the projection mapping P(es->set) is not used. Isn't it used in the latter? Where is the ECG-Text embedding space in the figure?
>
> * The projection mapping $P_{es \rightarrow set}$ is indeed used in our main proposed model, **sEHR-ECG-Text**. In this model, it is used in projecting ECG representations from the  fine-grained joint ECG-sEHR space $(\Omega_{es})$ to the  coarse-grained sEHR-ECG-Text space $(\Omega_{set})$.
> * To clarify, in bi-modal contrastive learning models, specifically **ECG-sEHR** and **ECG-Text**, which are distinct from the **sEHR-ECG-Text** model, there is no requirement for the $P_{es \rightarrow set}$ projection. These models involve only two modalities, and their representations can be learned jointly in a shared space by projecting modality-specific representations into that shared space, using a single projection head for each modality.
> * To address the confusion and provide greater clarity, we have removed this statement in caption of Figure 1 and have now presented separate figures for both ECG-sEHR (Figure 3(a)) and ECG-Text (Figure 3(b)) models in Appendix B.2.
> * Please note that in the **sEHR-ECG-Text** model, the contrastive loss between the ECG and text modalities is applied in the joint sEHR-ECG-Text space $(\Omega_{set})$. As a result, the ECG-Text space $(\Omega_{et})$ does not come into play in this model.
>
> >What's the motivation behind using the weighting scalars in the loss in equations 1 and 2?
>
> * The use of weighting scalars is a standard practice employed in multi-modal contrastive learning frameworks, as also discussed in ConVIRT [1]. The contrastive objective work bi-directionally, meaning that for each pair of modalities (e.g., ECG and sEHR), there are two loss components: one from ECG to sEHR and the other from sEHR to ECG. These scalars provide control over the relative significance of each loss term within the overall combined loss function.
>
> [1] Zhang et al., 2022, Contrastive
> learning of medical visual representations from paired images and text, In MLHC 2022.

---

> > ### Author Response · Authors · 2023-10-28
> > **Response to Reviewer V79p (2/5)**
> >
> > > What's the difference between equation 4 and equation 2?
> >
> > * Equations 2 and 4 serve different purposes. Equation 2 is utilized in the **sEHR-ECG-Text** model for contrastive learning between ECG and Text modalities in the coarse-grained sEHR-ECG-Text space $(\Omega_{set})$. This is used in conjunction with contrastive loss between ECG and sEHR modalities in fine-grained ECG-sEHR space $\Omega_{es}$) (Equation 1).
> > * In contrast, Equation 4 is used in the **ECG-Text** model. Note that sEHR modality is not involved in this model, so there is no requirement for two embedding spaces. The contrastive loss between ECG and Text modalities in this model is applied in the joint ECG-Text space $(\Omega_{et})$. See that $z$ in $v_{m,z}$ is $set$ in Equation 2 and $et$ in Equation 4. We provide model architectures for **ECG-sEHR** and **ECG-Text** models in Appendix B.2 for greater clarity.
> >
> > > Have the authors considered how this pre-training framework could generalize to the other input modalities (not just ECG)?
> >
> > * Indeed, this pre-training framework is designed to be versatile and can be readily extended to accommodate various input modalities beyond ECG. Note that we've already introduced modality-agnostic notations, which emphasize the flexibility of our approach. To apply this framework to different modalities, we would need a modality-specific encoder capable of encoding the specific data modality and a projection head to map the resulting representations into the desired embedding space for contrastive learning.
> > *  For instance, when dealing with echocardiogram videos, a 3D ResNet could be employed as the encoder.
> >
> > >Can you also provide a figure that clarifies how the modalities were linked with one another considering the varying timeframes / time differences between them?
> >
> > >Can you provide a figure that clarifies the difference between report and entity concatenation? It is unclear and the section is poorly written.
> >
> > We hope the following will provide clarity on how the modalities were linked with one another and difference between report and entity concatenation.
> > * Text modality includes clinical narratives, physician notes, patient histories, and other free-text entries, while structured EHR data (sEHR modality) adheres to standardized medical coding systems and terminologies, providing specific information about diagnoses, procedures, and more. To illustrate, let's consider two patient notes from the text modality within a specific time window around an ECG:
> >
> > |Patient note (text modality)|sEHR codes (sEHR modality)|Entities (text modality)|
> > | :------------------------- | :----------------------- | :----------------------- |
> > | **ST-segment elevation** in leads II, III, and aVF, suggestive of an acute inferior **myocardial infarction**. | I21.09 | ST-segment elevation, myocardial infarction |
> > | Chest X-ray indicates the presence of **pleural effusion** in the left lung. | J90 | pleural effusion |
> > * For the structured EHR (sEHR) modality, we obtain the I21.09 code for the first example and J90 for the second example. This results in a sequence *[I21.09, J90]* for the sEHR encoder. In contrast, for the text modality, we concatenate the patient notes: *'ST-segment elevation in leads II, III, and aVF, suggestive of an acute inferior myocardial infarction. Chest X-ray indicates the presence of pleural effusion in the left lung.'*, which we refer to as report concatenation. This sentence is processed through a tokenizer to obtain a sequence of tokens and is sent to the text encoder.
> > * It's crucial to note that the sEHR modality requires only 2 tokens, while the text modality needs 39 tokens, after tokenization using GatorTron's [2] tokenizer. This allows us to include information from a longer timeframe using sEHR modality within a given token limit. Consequently, we adopt a 1-year timeframe for sEHR and a 1-month timeframe for text around the ECG timestamp.
> > * To expand the timeline for the text modality, within the token limit (400 in our work, see Section 4.2.3), we extract only the entities from the patient notes: ST-segment elevation, myocardial infarction from example 1, and pleural effusion from example 2, and concatenate them. This results in *'ST-segment elevation myocardial infarction pleural effusion'*, which we refer to as entity concatenation. This way, token length is reduced while capturing relevant information.
> > * Please note that this example is just for illustrative purposes, and the real data contain much longer sequences in the text modality, and all sEHR codes are generated within health systems. We have re-worked on Section 4.2.3 to provide greater clarity regarding report and entity concatenation. We also include a figure illustrating how sEHR and text modalities are linked with ECG modality within specific time windows around ECG timestamp (Figure 2).
> >
> > [2] Yang et al., 2022, Gatortron: A large clinical language model to
> > unlock patient information from unstructured electronic health records.

---

> ### Author Response · Authors · 2023-10-28
> **Response to Reviewer V79p (3/5)**
>
> > The exact contributions of the work are unclear as currently listed in the abstract. There are many results in the main manuscript, however it is also unclear which framework leads to the best improvements (ECG-sEHR vs ECG-Text vs ECG-sEHR-Text vs ECG-MTL). It currently seems that all models perform better in different scenarios, which does not really clarify the strengths of the proposed approach.
>
> **On contributions**
> * Please note that we have removed ECG-MTL model from the paper for the reasons of relevance as stated in the first response (part 1) and re-worked on the abstract to make the contributions more clear. Our main contributions are proposing a set of three multi-modal contrastive learning models, i.e., sEHR-ECG-Text, ECG-sEHR, and ECG-Text and comparing those with current SOTA methods. These are also summarized in point wise manner at the end of Section 1.
>
> **On strengths of proposed approach**
> - **Comparison between ECG-sEHR and ECG-Text.** Please note that the ECG-sEHR model demonstrates superior performance on internal datasets, whereas the ECG-Text model excels on external datasets. We hypothesize that this difference arises from the fact that external datasets, such as PhysioNet2020 and Chapman, primarily comprise diseases that are commonly diagnosed from ECGs (i.e., arrhythmias, conduction blocks, etc.). These diagnoses are well-documented in the textual modality, particularly within ECG reports. As a result, the ECG-Text model performs best on these external datasets. In contrast, our internal datasets contain diseases for which labels are derived from EHR data. These labels are better captured by the sEHR modality. Therefore, the ECG-sEHR model outperforms the ECG-Text model when applied to internal datasets. We also discuss this in Section 4.5.1, paragraph 2 in the revised paper.
>
> - **Main strengths.** Our main proposed model, sEHR-ECG-Text, demonstrates strong generalization across all diseases i.e., coronary atherosclerosis, myocarditis, cardiac amyloidosis, PH, and AFib in NSR (see Table 2 and Table 3) and achieves comparable performance with ECG-Text model in case of external datasets, i.e., PhysioNet2020 and Chapman (see Table 1). It also offers the advantage of comparing different modalities for retrieval tasks. Therefore, leveraging multiple modalities is advantageous for learning the ECG patterns of complete medical landscape. We also discuss this in Section 4.5.1 (paragraph 1, point 5) in the revised paper.
>
> > It is unclear why they conduct those specific ablation studies. It would be interesting to explore the impact of using different modalities, rather than long/short ICD and report/entity concatenation as those are not the main contributions of the work.
>
> * Please note that our models, ECG-sEHR and ECG-Text, are themselves ablations that explore the impact of individual modalities.
> * As stated in the paper, that long ICD codes do not provide additional information over short ICD codes for large scale pre-training (Section 4.2.1), and entity concatenation is advantageous over report concatenation (Section 4.2.3). We believe it's valuable to present evidence for these claims, and that's why we included these ablations.
>
> > The proprietary dataset is private.
>
> * Please note that we also use two publicly available datasets: PhysioNet2020 and Chapman to evaluate all the models.
>
> > The code is not shared, I would highly recommend that they share their repository considering that work in this area is generally based on private models.
>
> * We appreciate your suggestion regarding code sharing. However, this work and its derivatives are actively utilized in commercial products. If it successfully passes through regulatory compliance, we will certainly consider sharing the code.
>
> >There is no statistical significance testing or confidence intervals provided. They also only report the average AUROC without any standard deviation.
>
> * We have now added standard deviation along with average AUROC. We also show the results within 1 standard deviation of the best result in bold.
> * Additionally, we conducted statistical significance testing of our generalized sEHR-ECG-Text model against each of the baselines on 10% training split using two-sided $t$-test by running each experiment 10 times using 10 different fractional splits (10%) derived from the original 100% training split. We observed significant improvement over all baselines with $p$-value less than 1e-5. We have also reported 95% confidence intervals. We present this in Appendix C (Statistical Testing) in the revised paper.
>
> >Have you considered reporting other relevant metrics? such as AUPRC
>
> * We have now added AUPRC performance. we present this in Appendix D (Table 13 and Table 14).
>
> >Why is the AUROC reported as a percentage and to two decimal places? It is typically reported as a three decimal figure.
>
> * Thank you for your observation. We have adjusted our reporting of AUROC to reflect the three decimal figures.

---

> ### Author Response · Authors · 2023-10-28
> **Response to Reviewer V79p (4/5)**
>
> >There is no evidence of any hyper parameter tuning. This can help clarify the robustness of the proposed approaches with respect to one another.
>
> * Due to budget and resource constraints, we did not perform hyperparameter tuning. However, to determine the learning rate, we conducted some basic tuning.
>
> >Can you provide a figure that clarifies the final downstream architectures for all models, including ECG-MTL?
>
> * Given the variations in designs and notations of different architectures, it was challenging to present a single comprehensive figure. However, we have addressed this by including detailed architectural diagrams for all models in Appendix B.2. Additionally, we have ensured that references to these diagrams are provided wherever the methodology is discussed in the main text.
>
> >Section 4.3.1 describes how the samples were labeled, can you create a table that summarizes the labeling procedure of each disease label. The text is difficult to follow.
>
> * As suggested, we have added a table showing the summary of the disease labeling procedure. Please see Table 8 in Appendix A.2.
>
> >The baselines are also not strong. Are there other multi-modal baselines that you can compare to? Did you perform any hyper parameter tuning for the baselines? What were the learning rates, etc.
>
> * As mentioned in Section 2, our study is the first instance of multi-modal models for learning ECG representations. There are no directly comparable multi-modal baselines available for ECG data.
> * We utilized a batch size/learning rate of 512/1e-3 for all baselines. Nevertheless, to ensure the robustness, we also explored, 256/1e-4 for CLOCS, 512/1e-1 for PCLR, following the recommendations outlined in the CLOCS [3] and PCLR [4] papers. These alternative settings yielded similar performance for all diseases. 3KG [5] employed the same hyperparameters as ours, with a batch size of 512 and a learning rate of 1e-3.
> * **3KG and CLOCS correctness**. The results achieved with 3KG by us are on par with the results reported by 3KG authors (0.872 vs. 0.890 on 100% split for PhysioNet2020 dataset). The results achieved with CLOCS by us are also on par with the results reported by CLOCS authors (0.946 vs. 0.959 on 100% split for Chapman dataset). We suspect these discrepancies arise from variations in the encoder and data distribution used for pre-training. 3KG employs the same dataset for both pre-training and evaluation, while we use our private dataset for pre-training. It's important to note that the PhysioNet2020 dataset contains Holter ECGs, which are out-of-distribution compared to our private dataset. This evidence ensures the correctness of our 3KG and CLOCS implementation. We discuss on this aspect in Section 4.5.1 (paragraph 1, point 4) in the revised document.
> * **PCLR correctness**. We evaluated the PCLR method using the [model](https://github.com/broadinstitute/ml4h/tree/master/model_zoo/PCLR) released by PCLR authors on internal datasets and achieved comparable performance with our implementation of PCLR. This [model](https://github.com/broadinstitute/ml4h/tree/master/model_zoo/PCLR) was trained on 3.2 million ECGs. The table below presents the comparison of linear classification performance (AUROC) between our PCLR model and the authors' released PCLR model. We suspect that the minor differences arise from variations in the ECG encoder. They use a larger model with 6.4 million parameters, while our model consists of 1 million parameters. This evidence ensures the correctness of our PCLR implementation.
>
> |Disease|Our PCLR model|Authors' released PCLR model|
> | :--------------------: | :----------------------: | :-----------------------------: |
> |Coronary Atherosclerosis |0.826|0.842|
> |Myocarditis|0.819|0.817|
> |Cardiac Amyloidosis|0.929|0.936|
> |Pulmonary Hypertension|0.908|0.903|
> |Low LVEF|0.936|0.937|
> |AFib in NSR|0.904|0.900|
> * We suspect that the main reason for the poor performance of ECG-only contrastive learning models on internal datasets is their exclusive dependence on ECGs for learning representations, i.e., comparing different instances of ECG data. This may not be sufficient to learn the complex patterns of various medical conditions. In contrast, our method aligns ECGs with EHR data, providing rich contextual information about a patient's health history, including diagnoses, procedures, medications, and more. This approach is beneficial for learning ECG patterns more effectively. We discuss this in Section 4.5.1 (paragraph 1, point 5) in the revised document.
>
> [3] Kiyasseh et al., 2021, Clocs: Contrastive learning of cardiac signals across space, time, and patients, ICML, 2020.
>
> [4] Diamant et al., 2022, Patient contrastive learning: A performant, expressive, and practical approach to electrocardiogram modeling, PLoS Computational Biology, 2022.
>
> [5] Gopal et al., 2021, 3kg: contrastive learning of 12-lead electrocardiograms using physiologically-inspired augmentations, ML4H, 2021.

---

> > ### Author Response · Authors · 2023-10-28
> > **Response to Reviewer V79p (5/5)**
> >
> > >There are many decisions that seem to be made based on heuristics, such as keeping ICD codes that are associated with at least 50 patients.
> >
> > * This is a mechanism to control the minimum frequency of sEHR codes that are included in the training process. This is to ensure we have enough data for effective training. It's important to note that a similar mechanism is employed in the [word2vec](https://radimrehurek.com/gensim/models/word2vec.html) algorithm.
> >
> > >There are typos and grammar mistakes.
> >
> > >consistency of capitalization in introducing abbreviations
> >
> >
> > * We apologize for these errors. We have now corrected it.
> >
> > >The paper is quite dense and extensive and hence the authors need to clarify certain elements of the text as described below. It mostly reads as a technical report, such that the methodological contributions / design elements are integrated with implementation details (hyper parameters, etc.)
> >
> > * We have provided implementation details including hyper parameters in Appendix B and have made references to it from the relevant sections in the main text where methodology is discussed (Sections 3.2, 3.3, and 3.4). This should enhance the clarity of our work.
> >
> >
> > >It's uncommon to refer to EHR as EHRs
> >
> > * In our manuscript, we utilized the term "EHRs" to refer to Electronic Health Records (plural form), aligning with common usage in the field. For further clarification and validation, we refer to official sources and established references to support our choice:
> >
> >   * [HealthIT.gov](https://www.healthit.gov/topic/health-it-and-health-information-exchange-basics/what-are-electronic-health-records-ehrs)
> >   * [National Institutes of Health (NIH)](https://www.nih.gov/sites/default/files/research-training/initiatives/pmi/opportunities-challenges-electronic-health-records.pdf)
> >
> > >The first paragraph in the introduction is too long and it's unclear what the main message is. This also applies to other sections in the text.
> >
> > * It's important to highlight that this study is first instance of multi-modal contrastive learning methods to learn ECG representations. Eventhough it is generally applied to medical image domain, it's relatively unexplored in ECG domain. The introduction emphasizes these key points.
> >
> > >"weekly" manner should be rephrased
> >
> > * Thank you for your suggestion. We have revised the phrasing to "time embeddings are constructed so that codes falling within non-overlapping 7-day windows share a common embedding."
> >
> > >All table and figure captions must be elaborated extensively.
> >
> > >The tables are referenced in random order. They also reference tables in the appendix in the main text without mentioning that they are supplementary.
> >
> > * Thank you for the suggestion, we have now elaborated all table and figure captions extensively.
> > * We have also reorganized the references to tables to follow a sequential order, and we've made it explicit that some tables are located in the appendix, rather than directly referencing them in the main text.

---

> ### Comment · Reviewer_V79p · 2023-10-31
>
> Thank you for addressing all my comments in (1/5), (3/5).
>
> In response 2/5, thank you for clarifying how the modalities are connected with one another. Can you please answer the below questions:
> - My understanding is that all the methods presented in the results table perform classification of ECG only (using a specific pre-training strategy). My main question is how your method learns from information that is acquired within +- 30 days / 1 year from the time of ECG collection, which would also increase the cost of pre-training. Can you provide the distribution of time between ECG and other modalities?
> - Can you also run another baseline where you amend the timelines, such as +- 15 days / 6 months (for the sake of example).
>
> In response 4/5, I still think that the lack of hyper parameter tuning is a major weakness.
>
> In response 5/5, there are still typos and grammar that needs to be fixed. Please revise thoroughly.

---

> > ### Author Response · Authors · 2023-11-08
> > **Response to Reviewer V79p**
> >
> > Dear reviewer, thank you for acknowledging the responses. We address the questions below.
> >
> > * The motivation behind pairing ECGs with EHR data from an extended timeframe during the pre-training is that, in the ECG labeling process for classification, we assign a positive label to all ECGs acquired within specific timeframes relative to the clinical diagnosis timestamp. These timeframes vary from 1 week to 1 year, depending on the disease, as outlined in Appendix A.2. This is a standard practice in ECG-related research. For further reference, we provide supporting evidence: coronary atherosclerosis [1] (see Figure 1 in the reference), cardiac amyloidosis [2][3], pulmonary hypertension [4], and structural heart diseases [5] (see Figure S1 in the [supplementary material](https://www.ahajournals.org/action/downloadSupplement?doi=10.1161%2FCIRCULATIONAHA.121.057869&file=CIRC_CIRCULATIONAHA-2021-057869_supp1.pdf) for greater clarity). These works suggest that ECGs obtained from an extended timeframe from the clinical diagnosis possess predictive capabilities for disease identification. As recommended by clinicians, we adopted 1-year time window to ensure comprehensive coverage for all diseases.
> >
> > * We have added time difference distribution plots between ECG and other modalities in Figure 3, Appendix A.1. The distributions are plotted by considering the number of ECGs associated with at least one medical code for the sEHR modality and at least one biomedical entity for the text modality, in one-week intervals.
> >
> > * Since the pre-training dataset is large and given our limited computational resources, the training of models with 15-day/6-month time frames is still ongoing and it appears that they cannot be completed before the deadline. We will share the results as soon as they are completed.
> >
> > * We apologize for the oversight. We have now fixed the typos and grammar in the latest version.
> >
> > References:
> >
> > [1] [Awasthi et al., 2023, Identification and risk stratification of coronary disease by artificial intelligence-enable	d ECG.](https://www.thelancet.com/journals/eclinm/article/PIIS2589-5370(23)00436-4/fulltext)
> >
> > [2] [Grogan et al., 2021, Artificial Intelligence–Enhanced Electrocardiogram for the Early Detection of Cardiac Amyloidosis.](https://pubmed.ncbi.nlm.nih.gov/34218880/)
> >
> > [3] [Harmon et al., 2023, Postdevelopment Performance and Validation of the Artificial Intelligence-Enhanced Electrocardiogram for Detection of Cardiac Amyloidosis](https://www.sciencedirect.com/science/article/pii/S2772963X23005744)
> >
> > [4] [Wagner et al., 2021, An Automated Screening Algorithm Using Electrocardiograms for Pulmonary Hypertension.](https://www.atsjournals.org/doi/epdf/10.1164/ajrccm-conference.2021.203.1_MeetingAbstracts.A1179?role=tab)
> >
> > [5] [Alvaro et al., 2022, rECHOmmend: An ECG-Based Machine Learning Approach for Identifying Patients at Increased Risk of Undiagnosed Structural Heart Disease Detectable by Echocardiography.](https://www.ahajournals.org/doi/epub/10.1161/CIRCULATIONAHA.121.057869)

---

> > > ### Author Response · Authors · 2023-11-15
> > > **Response to Reviewer V79p**
> > >
> > > Dear reviewer, please find the results below based on 15-day and 6-month time frames using the ECG-sEHR model. We report mean and standard deviation over 5 runs on 10% and 100% splits. Please note that we observe small performance gain when using long time frames (6 months/1 year) compared to short time frame (15 days).
> > >
> > >
> > >
> > > **Table 1: Linear classification performance (AUROC) on coronary atherosclerosis, myocarditis, and cardiac amyloidosis diseases.**
> > > | Time window | Coronary atherosclerosis (10%) | Coronary atherosclerosis (100%) | Myocarditis (10%) | Myocarditis (100%) | Cardiac amyloidosis (10%) | Cardiac amyloidosis (100%) |
> > > |-------------|-------------------------------|----------------------------------|-------------------|---------------------|---------------------------|---------------------------|
> > > | 15 days     | 0.834 $\pm$ 0.009                         | 0.889 $\pm$ 0.000                           | 0.853 $\pm$ 0.020            | 0.901$\pm$ 0.000               | 0.931 $\pm$ 0.003                    | 0.959   $\pm$ 0.000                   |
> > > | 6 months    | 0.842 $\pm$ 0.002                         | 0.892 $\pm$ 0.000                           | 0.851 $\pm$ 0.013            | 0.907 $\pm$ 0.001              | 0.936 $\pm$ 0.005                    | 0.961  $\pm$ 0.000                    |
> > > | 1 year      | 0.836 $\pm$ 0.011                        | 0.891  $\pm$ 0.000                          | 0.859 $\pm$ 0.011            | 0.896 $\pm$ 0.000              | 0.932  $\pm$ 0.006                   | 0.959    $\pm$ 0.000                  |
> > >
> > > **Table 2: Linear classification performance (AUROC) on pulmonary hypertension, low LVEF, and AFib in NSR diseases.**
> > > | Time window | Pulmonary hypertension (10%) | Pulmonary hypertension (100%) | Low LVEF (10%) | Low LVEF (100%) | AFib in NSR (10%) | AFib in NSR (100%) |
> > > |-------------|-----------------------------|--------------------------------|-----------------|-------------------|-------------------|---------------------|
> > > | 15 days     | 0.934  $\pm$ 0.001                     | 0.940 $\pm$ 0.000                         | 0.940 $\pm$ 0.001          | 0.950  $\pm$ 0.000           | 0.931$\pm$ 0.001             | 0.933 $\pm$ 0.000              |
> > > | 6 months    | 0.933 $\pm$ 0.001                       | 0.940 $\pm$ 0.000                         | 0.940 $\pm$ 0.001          | 0.951 $\pm$ 0.000            | 0.932$\pm$ 0.001             | 0.934 $\pm$ 0.000              |
> > > | 1 year      | 0.933 $\pm$ 0.001                      | 0.940 $\pm$ 0.000                         | 0.942 $\pm$ 0.001          | 0.951$\pm$ 0.000             | 0.930 $\pm$ 0.001            | 0.933   $\pm$ 0.000            |

---

### Decision · Action_Editor_5z3L · 2023-11-15

**Recommendation:** Accept as is

**Comment:**

The authors use existing techniques on a new dataset, and analyze performance results on both internal and external benchmarks.  Their analysis yields strong evidence that their multimodal approach can improve upon existing SSL-for-ECG techniques for producing predictors in low samples and match (or surpass) supervised baselines in larger data regimes.  While it is disappointing that the dataset that leads to these performance improvements is closed, it is understandable given the nature of the dataset. I believe these results will find an audience, perhaps, e.g., other researchers at health centers with access to their own ECGs and CHRs.  I definitely encourage following up with code for to aid in the reproduction of these results by such researchers.  Reviewer concerns were, for the most part, addressed.

**Audience:**

Yes, I strongly believe that there is an audience that will find these results of interest.

**Claims And Evidence:**

The authors present empirical results, highlighting the performance benefits of multimodal representations in low data regimes on both internal and external datasets.  They compare their multimodal representations to some standard ECG SSL methods as well as a supervised baseline.  Their interpretation of the results is sensible, and leaves room for uncertainty and further empirical investigation (e.g., their hypothesis that the ECG-text tends to do best on the PhysioNet2020 and Chapman datasets).